# Flexible read-aware genotype imputation from sequence using biobank sized reference panels

Zilong Li [1] ✉, Anders Albrechtsen [1] & Robert W. Davies [2,3] ✉

Inexpensive and accurate genotyping methods are essential to modern genomics and health risk prediction. Here we introduce QUILT2, a scalable and read-aware imputation method that can efficiently use biobank scale haplotype reference panels. This allows for fast and accurate imputation using short reads, as well as long reads (e.g. Oxford Nanopore Technologies (ONT) 1X, r2 = 0.937 at common SNPs), linked-reads and ancient DNA. In addition, QUILT2 contains a methodological innovation that is designed to enable imputation of the maternal and fetal genome using cell free non-invasive prenatal testing (NIPT) data. Using a UK Biobank reference panel and simulated NIPT data, we see accurate imputation of the mother (0.25X, r2 = 0.966, common SNPs) and modest imputation of the fetus (0.25X, r2 = 0.465, fetal fraction of 10%) at low coverage, with fetal imputation accuracy rising with coverage (4.0X, fetal r2 = 0.894). We show using simulated data that this could enable both GWAS and PRS for the mother and fetus, which could create clinical opportunities, and if phenotypes can be collected alongside clinical NIPT, the potential for large GWAS.

Low-coverage whole genome sequencing (lc-WGS) was originally demonstrated to be a cost-effective alternative to DNA genotyping arrays for genotyping of modern genomes in 2012[1]. However, early use of lc-WGS used statistical methods for phasing and imputation originally designed for microarrays[2]. Subsequently, efficient dedicated methods for imputation from lc-WGS were designed, which could operate both with or without reference panels, which were generally shown to be more powerful than approaches using arrays[3–6]. Additionally, imputation with reference panels has been demonstrated as a reliable approach for ancient DNA (aDNA) studies[7]. Leveraging large reference panels, aDNA imputation can be further improved[6]. Furthermore, human genetic studies have begun using long-read sequencing more and more for investigating disease-associated variants, and a strategy of mixing high and low coverages is favored for population-scale study[8].

Previously, we developed QUILT for rapid genotype imputation of lc-WGS data with a reference panel[4]. Around the same time, Rubinacci et al. introduced GLIMPSE, another method for genotype imputation[4,5]. Both QUILT and GLIMPSE share a core two-stage imputation framework, where per-sample imputation leverages a small set of conditioning reference haplotypes. For haplotype matching, QUILT employs the Li and Stephens model[9], which is more robust, whereas GLIMPSE relies on the positional Burrows-Wheeler transform[10] (PBWT), which offers computational efficiency. QUILT and GLIMPSE further differ in how they treat input data, with GLIMPSE using per-SNP genotype likelihoods, while QUILT uses and phases the sequencing reads directly, which is preferable for linked reads or long-read data. Overall, QUILT is at least as accurate as GLIMPSE and more robust, however GLIMPSE tends to be faster[5]. Since the publication of these methods, reference panels have expanded significantly in both the number of haplotypes as well as variant density, for example many studies now use the UK Biobank whole-genome sequencing (UKB WGS)[11]. To accommodate these growing panel sizes, the authors of GLIMPSE developed GLIMPSE2[6]. GLIMPSE2 focuses on improving speed and facilitating biobank sized reference panels by incorporating more sparse data

[1]Computational and RNA Biology, Department of Biology, University of Copenhagen, Copenhagen, Denmark. [2]Department of Statistics, University of Oxford, Oxford, UK. [3]Genomics Ltd, Oxford, UK. ✉e-mail: zilong.dk@gmail.com; robertwilliamdavies@gmail.com

structures for both speed and memory efficiency, as well as the introduction of a dedicated version of PBWT, termed sparse PBWT.

For human GWAS, studies have estimated sample sizes in the millions are required to robustly detect signals that explain most of the heritability underlying common traits and diseases[12]. This problem is only exacerbated when we consider the need for GWAS in many different populations, to enable population specific effect size estimation, and to help realize the promise of precision medicine for all[13]. One option is to utilize DNA and phenotypes from pregnant women. Electronic health records (EHR) usually store rich phenotypes for pregnant women, which could include quantitative traits like height, weight, hyperglycaemia[14], as well as socioeconomic traits like educational attainment, as well as disorders with age of onset before maternal age or during pregnancy, such as neuropsychiatric conditions like schizophrenia, as well as gestational diabetes mellitus[15] and intrahepatic cholestasis of pregnancy[16]. A natural way to collect DNA from pregnant women is using cell free DNA from non-invasive prenatal testing (NIPT), which is a sensitive and specific screening technology which tests for the common fetal aneuploidies of trisomies 13, 18 and 21, and that, according to the American Society of Obstetricians and Gynecologists (ACOG, 2020), should be offered for all pregnancies regardless of risk. In some countries, NIPT is routinely offered to all pregnant women, with recent uptake rates at around half the population or more, with an uptake rate of 79% and 46% for Belgium and Netherlands respectively[17]. In 2018, it was suggested that NIPT had already been performed on ten million pregnant Chinese women[18]. Further, NIPT has already been used as a successful means of conducting GWAS, with NIPT based GWAS discovering novel associations for intrahepatic cholestasis of pregnancy[16], gestational diabetes mellitus[15], metabolites[19], glycemic traits[20] and neonatal phenotypes[21]. Separately, recent studies proposed high-resolution noninvasive prenatal screening to investigate the entire fetal exome from circulating cfDNA by whole exome sequencing (WES)[22,23], which usually offers genome-wide off-target reads for non-coding regions for 1×–2× coverage[24]. The increased benefit of such a test over screening only for aneuploidies should increase the rate of adoption of such tests worldwide. However, state-of-the-art imputation methods for lc-WGS are designed for diploid samples, and there is no dedicated method designed for NIPT data taking both the fetus DNA and mother DNA into account. Given different timepoints of sequencing cfDNA in the blood of pregnant women, the fetal fraction (FF) varies between 4% and 50%[23], and at the most commonly assayed gestational ages of 10 to 20 weeks, is about 10–15%[25]. Without dedicated imputation methods for NIPT data, imputation accuracy would be negatively correlated with fetal fraction in cfDNA[26]. As such, the underlying data suggests that imputation from NIPT can be improved, and the scale of NIPT suggests it would be very valuable[27,28]. Furthermore, imputing fetal genomes before childbirth may facilitate health risk management through polygenic risk scores (PRS) for perinatal traits in clinical practice.

In this study, we present QUILT2, a novel scalable method for rapid phasing and imputation from lc-WGS reads and cfDNA using very large haplotype reference panels. QUILT2 contains three key innovations, two technical and one methodological, compared to QUILT (or QUILT1). First, we introduce a memory efficient version of the positional Burrows-Wheeler transform (PBWT)[10], which we call the multi-symbol PBWT (msPBWT). QUILT2 uses msPBWT in the imputation process to find haplotypes in the haplotype reference panel that share long matches to imputed haplotypes with constant computational complexity, and with a very low memory footprint. Second, we introduce a two stage imputation process, where we first sample read labels and find an optimal subset of the haplotype reference panel using information at common SNPs, and then use these to initialize a final imputation at all SNPs. This both speeds up imputation as well as decreases the amount of RAM required. Finally, we introduce a novel

methodological innovation to QUILT2, which we call the QUILT2-nipt method, which assumes the observed sequencing reads are from the three haplotypes present in NIPT data, and which imputes both the maternal and fetal genomes using DNA from pregnant women. In what follows, we evaluated the accuracy, performance and scalability of QUILT2 for various data types on diploid samples. In addition, we evaluated the accuracy of imputing the maternal and fetal genomes from simulated NIPT data. Finally, we show how this data can enable both GWAS and PRS, in both the mother and the fetus.

## Results
### Model overview

QUILT2 is based upon and improves our previous model QUILT1 for diploid genotype imputation, which uses an iterative approach, as follows. First, Gibbs sampling is performed to generate a posterior estimate of the read labels (an estimate of what haplotype each read comes from) given the sequencing reads and the haplotypes in a small haplotype reference panel using the Li and Stephens algorithm[9]. Second, the small haplotype reference panel is updated based on the current estimate of the haploid dosages, again using Li and Stephens. Due to the second stage in haplotype selection, QUILT1 has linear time complexity in the number of haplotypes and variants in the full reference panel. For RAM and speed reasons, in QUILT1, we disable transitions except between every 32nd pair of SNPs, and store haplotypes as 32 bit integers (or a further representation of this). We refer to 32 consecutive SNPs as a grid.

With QUILT2, we first addressed the scaling of QUILT1. The use of PBWT is common for haplotype identification in genotype phasing and imputation[5,6,29–31], including with GLIMPSE and GLIMPSE2, and derivatives exist that work on non-binary symbols, such as multi-allelic and Syllable PBWT[32,33]. We similarly developed our own analogue of PBWT, which we call the multi-symbol PBWT (msPBWT), which naturally operates on the symbols in the grids used in QUILT1 and QUILT2 (Supplementary Fig. 1). With msPBWT, and using pre-computed indices of the reference panel, we use an algorithm to identify haplotypes in the full reference panel that have long identical matches against a target imputed haplotype with computational time independent to the panel size. Next, we note with WGS derived reference panels, the vast majority of SNPs are very rare, and as such, will slow down methods that are linear in the number of SNPs. In addition, these rare SNPs are of marginal importance in determining which reference haplotype to copy from when there are centimorgan or longer haplotype matches between target and reference haplotype. As such, several methods for genotype phasing and imputation differentiate between common and rare SNPs, for both speed and RAM reasons, including GLIMPSE2 and other methods[29,31,34]. For QUILT2, we introduce such a two step process, where we iteratively perform imputation using only common SNPs and the reads that intersect them, to get an optimal subset of the full reference panel (Supplementary Fig. 2). We then load all reads at all sites, initialize their phase according to the current best estimate of the underlying haplotypes, and perform a final round of Gibbs sampling, to get final read labels, and from them haplotype phase and genotype dosage. Finally, methodologically, QUILT2 introduces a novel method for jointly imputing the mother and fetus utilizing the lc-WGS cfDNA data from NIPT (Fig. 1). This novel method includes changes to the mathematics as compared to the original diploid model, for instance the probability of the reads given the read labels and the haplotypes, as well as the probability of the read labels given the fetal fraction, to accommodate the three haplotypes present in different frequencies in cfDNA. Further, the change in the prior on read labels, which goes from uniform to unequal and dictated by the fetal fraction, was incorporated throughout the modeling, which necessitated substantial changes to the heuristics to minimize the Gibbs sampling getting stuck in local extrema. Further details, including for msPBWT, common variant initialized imputation, NIPT, as well heuristics, are described in Methods and Supplementary Note.

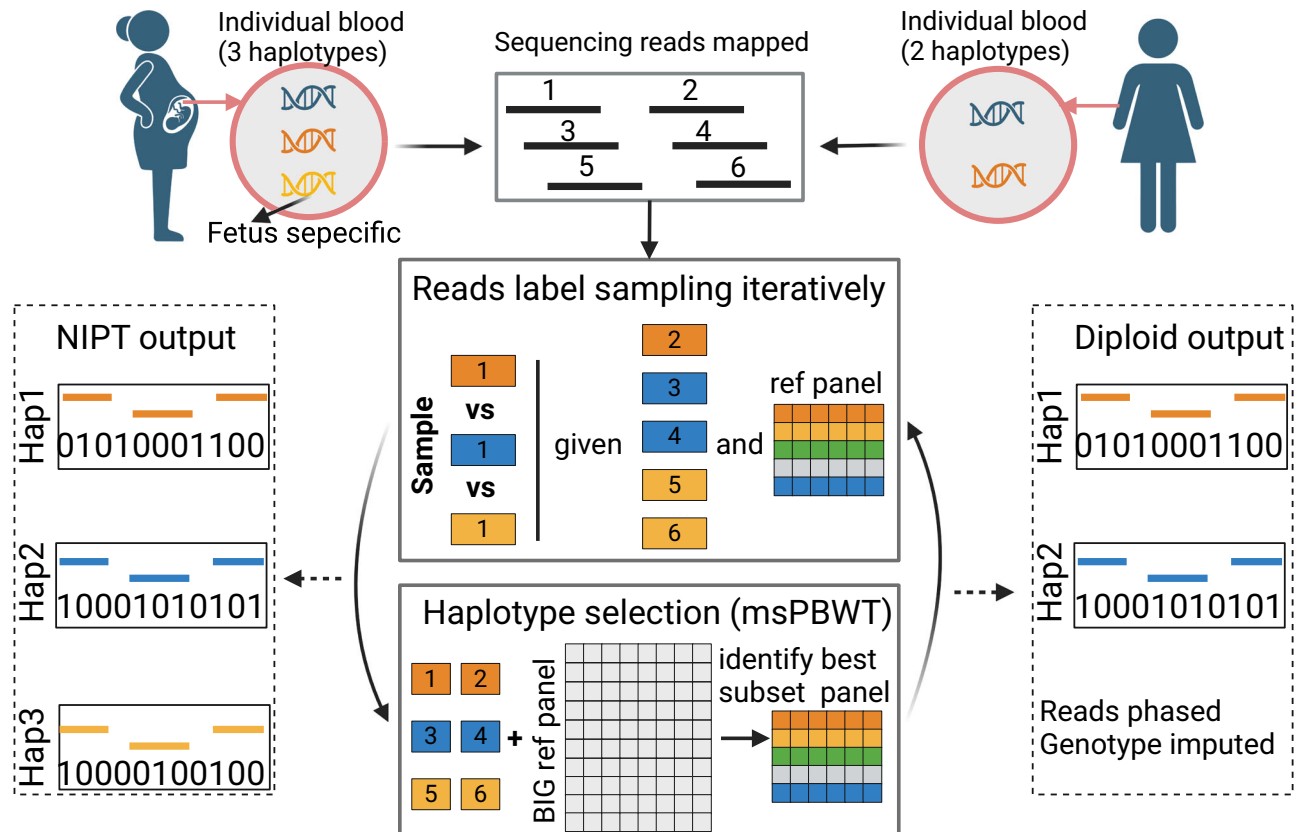

**Fig. 1 | Schematic of the QUILT2 model.** The model has two modes, called diploid and nipt mode, where the obvious difference is the number of underlying haplotypes modeled (2 vs 3). For both modes, QUILT2 first partitions the reads into different sets using Gibbs sampling and performs haploid imputation using a subset of reference haplotypes, and then obtains a new subset of the reference panel using msPBWT and the full reference panel. These two procedures are done iteratively several times to reach the final optimal phasing and imputation. Through this process, the input sequencing reads (unlabeled, black) are phased into different haplotypes (labeled, orange, blue, yellow), and genotypes are imputed and phased. More details about the msPBWT algorithm are shown in Supplementary Fig. 1. Details about the iterative approach using common and all SNPs, which is not shown in this schematic, is shown in Supplementary Fig. 2. This figure was made using BioRender.com.

## Diploid imputation performance

In the benchmarking of diploid imputation performance, we predominantly analyzed European samples in the 1000 Genomes. To reduce the computational burden, we analysed only chromosome 20, and unless noted, focus on the analysis with three different reference panels: the 1000 Genome Project[35], $N$ (haps)=5,008, $M$ (SNPs) =2,115,074), the Haplotype Reference Consortium[36] (HRC, $N$ = 54,330, $M$ = 884,932), and the UK Biobank 200 K WGS genomes[11,37] (UKB, $N$ = 400,022, $M$ = 14,075,021). We first investigated the imputation performance of QUILT2 and GLIMPSE2 for diploid genomes using various data types with multiple sequencing coverages. Throughout the results, unless noted, we stratified accuracy of European samples by allele frequencies from the separate Genome Aggregation Database (gnomAD v3.1.2 Non-Finish European). As exemplars, we used very rare to refer to SNPs with MAF of 0.01-0.02%, rare to refer to SNPs with MAF of 0.1–0.2%, and common to refer to SNPs with MAF of 10-20%.

We first considered imputation using short read Illumina data from the CEU population of the 1000 Genomes. As shown in Fig. 2a and Supplementary Fig. 3, QUILT2 was more accurate than GLIMPSE2 at sequencing coverage ≤0.5× regardless of the reference panel, while GLIMPSE2 was more accurate than QUILT2 for rare variants at coverage ≥2.0×. For higher coverage samples ≥1.0×, at common SNPs, accuracy was similar between the panels and between the methods, while at rarer SNP, far more rare SNPs were imputed, and more accurately imputed with the larger panels (Supplementary Fig. 3, Supplementary Tables 1, 2). For imputation of modern samples with smaller

panels (KGP and HRC), we found that QUILT2 is less robust and accurate than QUILT1 and GLIMPSE2 for rare variants with default settings (Supplementary Fig. 3). For QUILT2 with the big UKB panel, we also evaluated the effect of the msPBWT haplotype selection versus the Li and Stephens approach of QUILT1, and note the msPBWT approach generates slightly more accurate results (Supplementary Fig. 4). In terms of phasing, the larger UKB panel substantially improved QUILT2 phasing accuracy compared to the smaller panels (Supplementary Fig. 5,6), and mirroring the imputation results, phasing with QUILT2 was more accurate than GLIMPSE2 particularly for lower coverages.

We also assessed the imputation performance on aDNA from Afanasievo culture (~4.6 kya) with three reference panels. We followed the Mapache[38] pipeline used in the GLIMPSE2 paper, which implements the practice by Mota et al. 2023[7]. Interestingly, QUILT2 outperformed GLIMPSE2 across all frequency variants using KGP and HRC panel, while both performed similarly using the UKB panel (Fig. 2b, Supplementary Fig. 7). Similar to results for modern DNA, the imputation of aDNA showed similar accuracy for common variants across reference panels, while the UKB panel again improved the performance of rare variants.

Lastly, we evaluated the imputation accuracy using long sequencing reads from Oxford Nanopore Technologies (ONT) by calculating both the $r^2$ at variant level and F1-score at sample level. To investigate the gain of using the UKB panel for long-read imputation, we included QUILT1 with HRC for comparison. We found that QUILT1-HRC

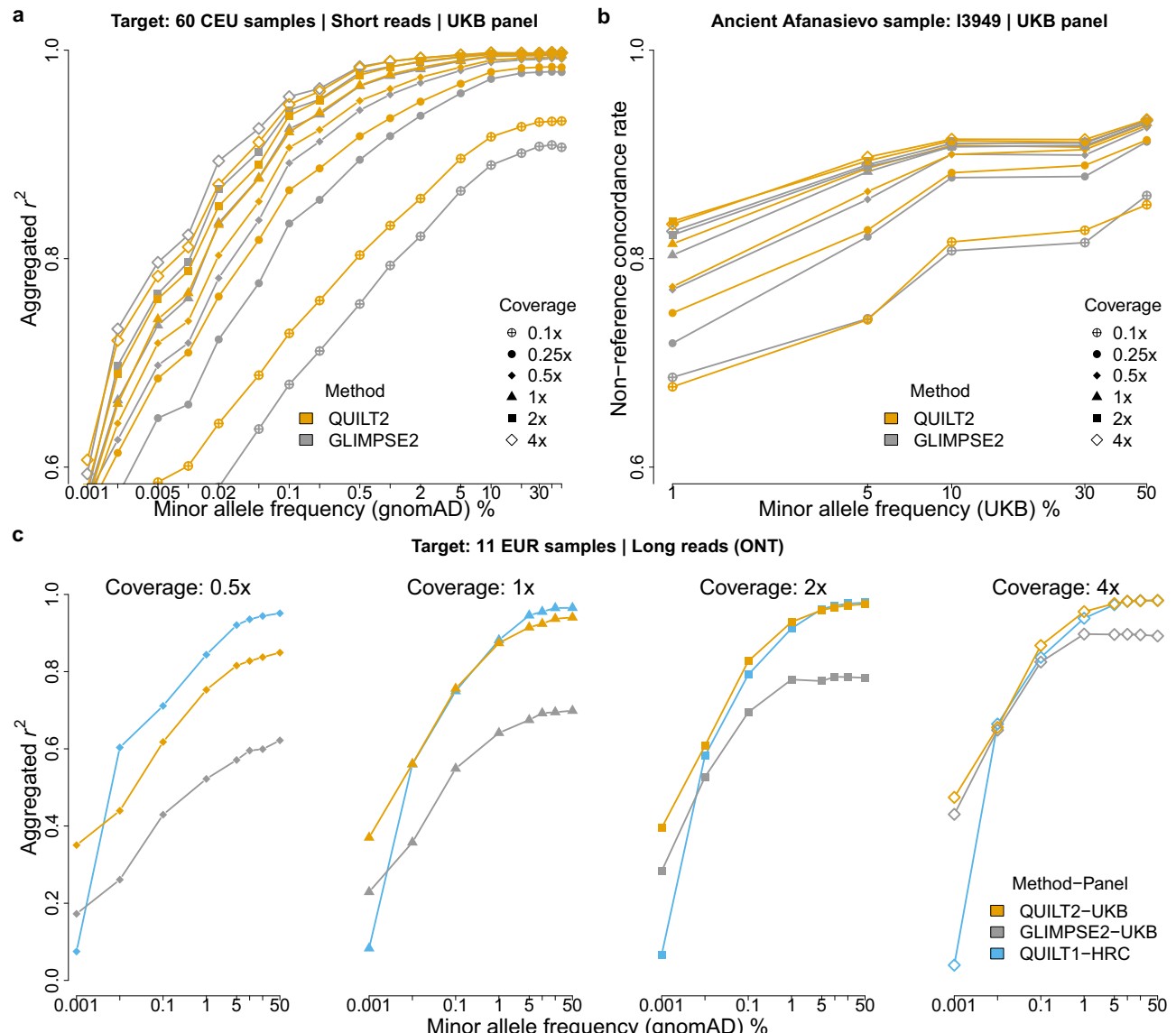

**Fig. 2 | Diploid imputation on various data types. a**, **b** Imputation performance for short-read data on modern European samples (**a**) and ancient (~4.6 kya) Afanasievo samples (**b**) across different reference panels and sequencing coverages for QUILT2 (orange) and GLIMPSE2 (gray). Accuracy, measured as the Pearson correlation coefficient ($r^2$) between imputed dosages and true genotypes or the non-reference concordance rate between imputed genotypes and true genotypes, is stratified by MAF. **c** Imputation performance of difference methods for long-read data on modern European samples. The HRC panel is used for QUILT1 since it is not scalable to the UKB panel.

performed better than QUILT2-UKB for coverage <1.0x but with fewer variants imputed (Fig. 2c). Additionally, we evaluated QUILT2 and GLIMPSE2, the inexpensive and scalable methods that can use the UKB panel. Across all coverages and reference panels, QUILT2 was substantially more accurate compared to GLIMPSE2 for long reads (Fig. 2c, Supplementary Table 3), e.g., common $r^2 = 0.937$ vs $r^2 = 0.695$ at 1× coverage.

## Scalability and computational cost

We next evaluated the computational cost and scalability of QUILT2 along with GLIMPSE2 for both speed and memory usage, varying both the size of the haplotype reference panel, as well as the number of target samples (with 1x coverage) to run. As shown in Fig. 3a, QUILT2 has approximately constant runtime with respect to haplotype reference panel size, and the per-sample runtime is not affected appreciably by the input sample size. By contrast, the per-sample runtime of GLIMPSE2 is affected by the number of samples run, and only shows approximately constant computational complexity with respect to

haplotype reference panel size for larger number of input samples ($N > 100$) but not for smaller size, as has been previously noted by the authors of GLIMPSE2[6]. In the extreme case, when only imputing one sample at a time with around 1 million haplotypes in the panel, QUILT2 would be an order of magnitude faster than GLIMPSE2. Further, QUILT2 with both msPBWT and the two-stage imputation enabled was ~3× faster and used ~4× less RAM, than QUILT2 with only msPBWT (Supplementary Fig. 8). Comparing QUILT2 and GLIMPSE2 for RAM, we see that both approaches show linear increases in RAM with reference panel size (Fig. 3a). Given around 1 million reference haplotypes and an average chunk size of around 5 Mb and 500Kb buffer to impute on chromosome 20, the maximum RAM for both is less than 10 GBs. RAM usage is approximately constant for GLIMPSE2 with sample size, whereas it increases in QUILT2 (Fig. 3). In addition, QUILT2 also runs faster for lower coverage data (Supplementary Figs. 8, 9). Since the UKB panel is only accessible on the UK Biobank research platform (RAP), we also assessed the cost of imputing whole genomes on the RAP. Since both QUILT2 and GLIMPSE2 do not borrow information

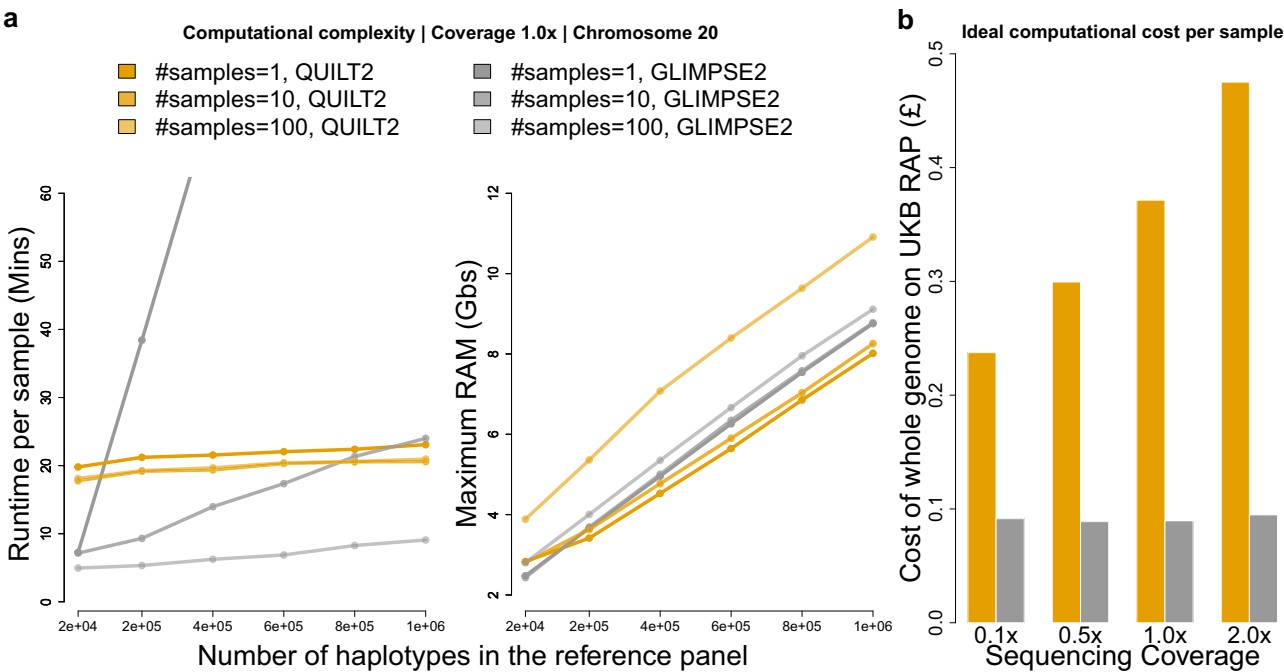

**Fig. 3 | Scalability and computational cost.** The benchmarking for QUILT2 (orange) and GLIMPSE2 (gray) was performed on virtual machines of type mem2_ssd1_v2_x4 with 4 CPU cores and 16 GBs memory on the RAP. Regarding the scalability (**a**) six different sizes of reference panel and three different sizes of input samples were tested. The number of SNPs in the reference panels varies, reflecting the removal of SNPs that become monomorphic in a subset of the panel. Time and RAM were measured for each chunk on chr20 by /usr/bin/time -v and one thread was used here. **b** for estimating the real cost on RAP, both QUILT2 and GLIMPSE2 were run in parallel with 4 threads to maximize the usage of the machine with 4 cores. We used jobs with $N = 32$ for QUILT2 and $N = 100$ for GLIMPSE2, as small jobs are less expensive but GLIMPSE2 achieves optional per-sample run time around $N = 100$ samples. The cost of the whole genome was estimated based on the bill of imputing chr20 by multiplying a factor of 34.48.

across samples, and as the cheapest way to run jobs on the RAP is using low or normal priority jobs which can be interrupted[6], it is advantageous to impute large datasets using small batches, and then to combine results afterwards. In their previous work, the authors of GLIMPSE2 showed that optimal per-sample runtimes were achieved at approximately 100 samples[6]. Comparing per-sample costs for QUILT2 (using $N = 32$) and GLIMPSE2 (using $N = 100$), we see both methods are inexpensive, though QUILT2 (£0.237, £0.299, £0.371 and £0.475 for 0.1×, 0.5×, 1.0× and 2.0×, respectively) is more expensive than GLIMPSE2 (approximately £0.09 for all coverages). We note these are idealized estimates, and real costs will depend on prices at run time, server congestion, etc, and furthermore, that default parameter settings for QUILT2 favour accuracy over speed.

## Imputation performance on simulated NIPT data and one real NIPT sample

We next investigated the QUILT2-nipt method for maternal and fetal genome imputation. We simulated 30 NIPT samples by mixing the sequencing reads of the mother and child in the 30 CEU trios from KGP (Methods). Here, we focus on NIPT imputation using the UKB panel across different sequencing coverages and various fetal fractions (FF). Firstly, we compared the QUILT2-nipt method with the existing diploid method (i.e. QUILT2-diploid) for maternal genotype imputation. As shown in Fig. 4a and Supplementary Table 4, given FF = 0.1, there is a clear gain of accuracy in QUILT2-nipt over QUILT2-diploid (0.25×, very rare $r^2 = 0.724$ vs 0.695; 1.0×, very rare $r^2 = 0.831$ vs 0.799; 2.0×, very rare $r^2 = 0.851$ vs 0.839). At higher sequencing coverage, for higher fetal fractions, QUILT2-diploid performed increasingly poorly, while QUILT2-nipt still maintained high accuracy for common variants as expected (e.g. 4.0×, FF = 0.2, common $r^2 = 0.994$, 0.853 for QUILT2-nipt and QUILT2-diploid, respectively). In addition to simulations, we also confirmed the effectiveness and improvements of QUILT2-nipt

over QUILT2-diploid for maternal genotype imputation with one real NIPT sample (Supplementary Fig. 10).

Next, we investigated the performance of QUILT2-nipt on fetal genotype imputation where there are no other approaches for comparison. As shown in Fig. 4b and Supplementary Table 5, both increasing fetal fraction, as well as sequencing coverage, contribute to improved accuracy, at both rare and common SNPs. For instance, at 0.25× coverage, accuracies at common SNPs improve with higher FF ($r^2 = 0.465$, 0.657, 0.783 for FF = 0.1, 0.2, 0.3, respectively), and they improve further at higher coverage with 4.0x ($r^2 = 0.894$, 0.944 and 0.955, same FFs). However, we do not observe a gain in accuracy when the sequencing coverage exceeds 4.0×, assuming FF = 0.3 (Supplementary Table 5). We note that $r^2 = 0.955$ appears to be the upper limit for fetal genotype imputation using the UKB panel, versus 0.92 and 0.90 for the HRC and KGP panels, respectively (Supplementary Fig. 11). This contrasts with the performance for the mother, where the upper limit remains 1.0 regardless of the reference panel. This improved performance for the larger panel is driven by the improved phasing capabilities offered by the larger reference panel (Supplementary Fig. 5), enabled by msPBWT within QUILT2, which is especially important for phasing the reads from the paternally transmitted haplotype. In terms of the run time, where mother and fetus are imputed together, we note that the QUILT2-nipt method is about 30% slower than the QUILT2-diploid method and imputing only common variants can reduce both the runtime and memory substantially (Supplementary Fig. 9).

## Power in association testing and prediction with simulated NIPT data

Finally, we explored the usability of imputed results from QUILT2-nipt for both GWAS and PRS. To create realistic NIPT sequencing samples, we used UK Biobank samples, for which we have real phenotypic data, and used parent offspring pairs to represent the mother and

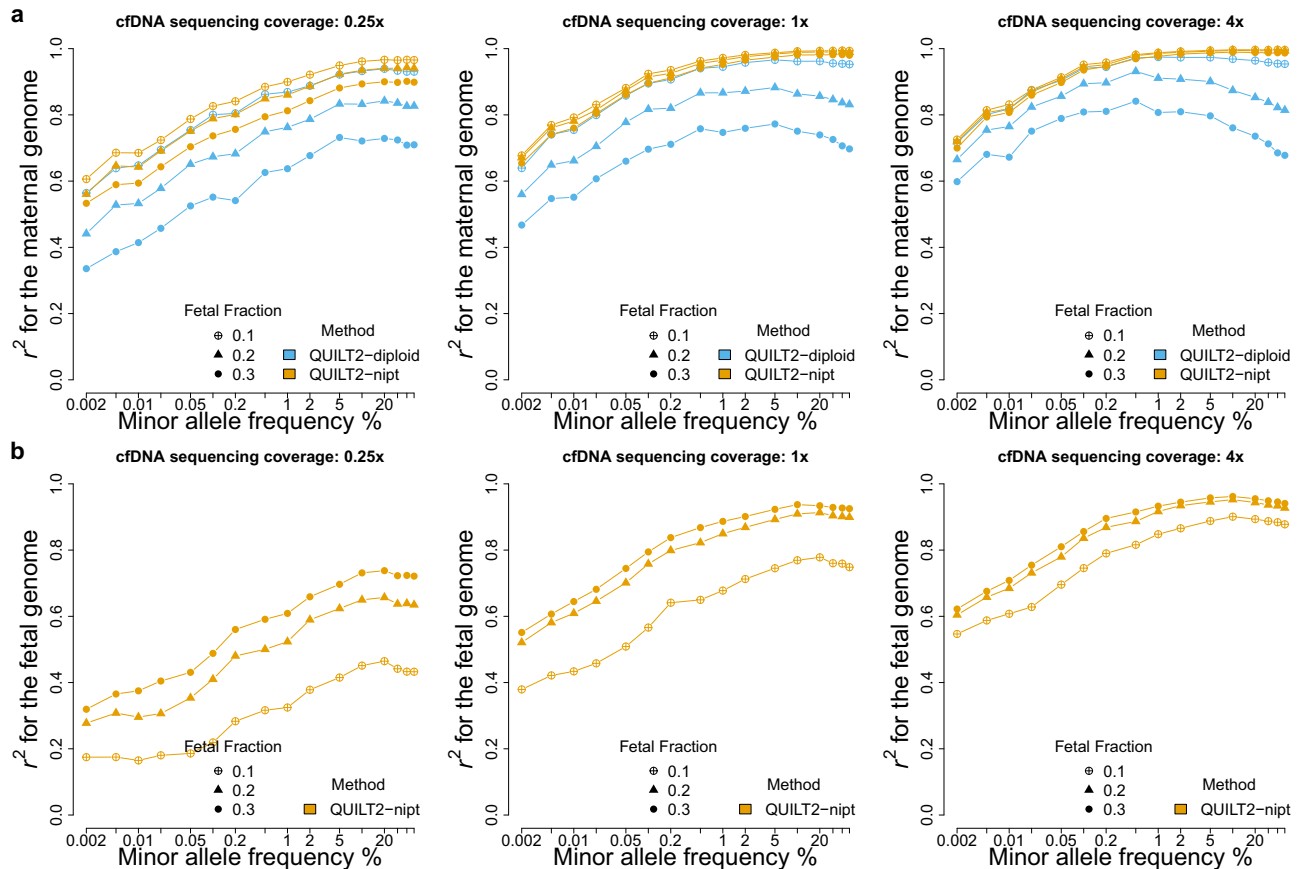

**Fig. 4 | Maternal and fetal genome imputation from simulated NIPT data.** The simulated 30 NIPT samples from CEU trios were analyzed. **a** imputation performance for the maternal genome across different NIPT sequencing coverages and various fetal fractions. Both QUILT2-diploid and QUILT2-nipt are able to generate the imputed genotypes of the mothers. **b** imputation performance for the fetal genome, where only QUILT2-nipt can impute the fetal genome.

the fetus ("Methods"). We used 11,028 parent-offspring pairs and simulated mean sequencing coverages of 0.25×, 1.0×, and 2.0×, with fetal fractions being simulated with mean 10% and following a normal distribution, with range from 5% to 15% (Supplementary Fig. 12). We conducted association tests on chromosome 1 for 25 quantitative traits using a linear mixed model on these 11,028 individuals (Methods). We only tested quantitative traits but expect binary traits to show broadly similar results.

We compared the GWAS association findings using QUILT2-nipt imputed maternal and fetal genotype dosages to a baseline that used imputed genotype dosages from arrays. With the QUILT2-nipt imputed data, we found that at common SNPs the $r^2$ for maternal genotypes were 0.973, 0.994, 1.000 at 0.25×, 1.0×, 2.0× coverage respectively, and the common $r^2$ for the fetal genotypes were 0.480, 0.75 and 0.82 (Fig. 5a). GWAS summary statistics based on different imputed datasets did not show signs of inflation of test statistics as noted by Q-Q plots (Supplementary Fig. 13). Next, we assessed the power of the different coverages and approaches, by measuring how many of 615 independent GWAS signals on chromosome 1 found using the entire UK Biobank participants could be identified ($p < 0.05$) using the imputed datasets of 11,028 simulated NIPT samples (Methods). Using the array-based imputation, 190 associations were found, while for QUILT2-nipt maternal imputation, 190 (100% estimated relative power), 188 (99%) and 181 (95%) were found at 2.0×, 1.0×, and 0.25×, respectively. For QUILT2-nipt fetal imputation, 163 (86%), 159 (84%) and 106 (56%) of loci were validated for 2.0×, 1.0×, and 0.25× data respectively (Fig. 5b). Notably, even though the common $r^2$ for fetal genotypes was only 0.82 at 2.0× with 10% FF, we still obtained a relative power of 86% in the GWAS. This is further validated in an error-prone

setting where the estimates of fetal fraction are measured with uncertainty and thus the prior for QUILT2-nipt method is mis-specified (Supplementary Fig. 14).

In addition to GWAS, the fetal imputed genotypes from QUILT2-nipt can be used to generate fetal polygenic risk scores, which can have potential clinical value for perinatal traits. We therefore evaluated the performance of PRS using QUILT2-nipt imputed fetal genotype dosages for 1000 individuals (Methods). We examined 21 traits with incremental $r^2 > 0.1$ (except for the waist-hip ratio) in unrelated white British Europeans[39], which is the difference in $r^2$ between the full prediction model (including PRS) versus the model with covariates alone (age, sex, age×sex, assessment center, genotyping array, 10 PCs). Using the reported PRS weights of each SNP, we calculated the PRS for each individual given different imputed datasets, that is the array-imputed, as well as NIPT imputed from 2.0×, 1.0×, and 0.25× coverage. We compared the incremental $r^2$ of the NIPT imputed datasets to the array imputed datasets, referring to this comparison as the relative incremental $r^2$ (Methods). As shown in Fig. 5c and Supplementary Fig. 15, with effective sequencing coverage for the mother at least 0.2× (e.g. 0.25×, 10% FF), we observed a relative incremental $r^2 > 0.967$. For the fetal PRS, while at low effective sequencing coverage, relative incremental $r^2$ is modest (e.g. 0.25×, 10% FF, giving 0.025× coverage, $r^2 = 0.494$), at high effective sequencing coverage (e.g. 2.0×, 20% FF, giving 0.4×), we observed a high relative incremental $r^2 > 0.935$. As expected, the PRS performance is aligned with the imputation accuracy.

## Discussion

Here, we introduced QUILT2, a novel method for imputation from lc-WGS. QUILT2 includes both technical improvements, which increase

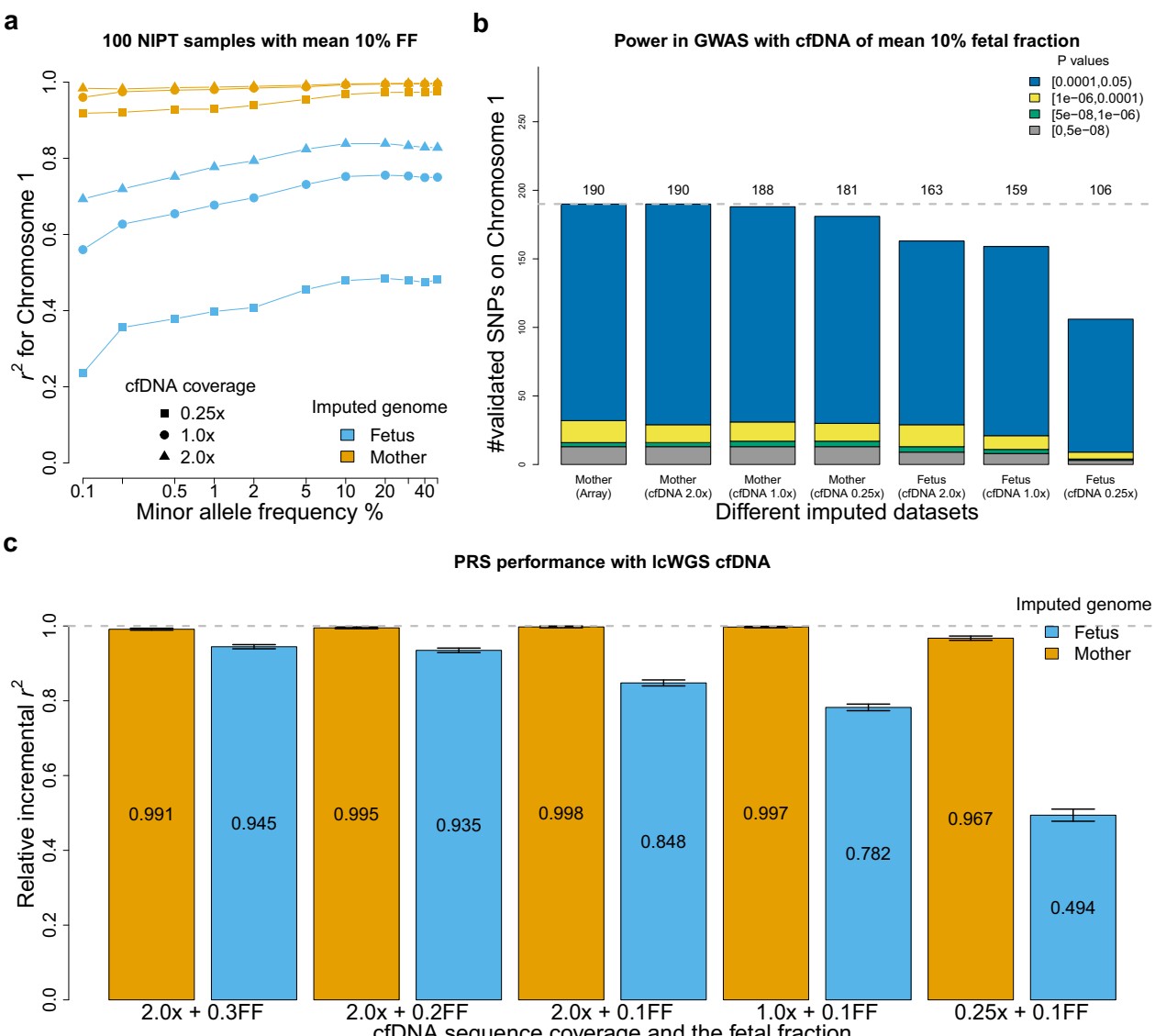

**Fig. 5 | Power in GWAS and PRS with simulated NIPT data. a** imputation accuracy of both maternal and fetal genotypes on chromosome 1 using the UKB panel for different cfDNA sequencing coverage with mean 10% fetal fraction. **b** Power in GWAS with both imputed mother and fetus genotype, compared the array-imputed data. Here we compare the number of SNPs that achieve p-values in a given range (linear mixed model association test, *p*-values not adjusted for multiple testing) using the target imputed datasets of 11,028 samples. **c** PRS performance for $N = 21$ traits with the whole imputed fetal and maternal genome from cfDNA with various sequencing coverage and fetal fractions. The mean value of relative incremental $r^2$ is shown at the middle of each bar. The error bar represents the standard error obtained from a one sample Student's *t*.test on the relative incremental $r^2$ of $N = 21$ traits.

the speed and scalability versus our previously released method QUILT1, as well as a methodological improvement to allow it to impute both mother and fetus from NIPT data. For diploid samples, the WGS-derived UKB panel showed that QUILT2 is as accurate as QUILT1 but significantly faster, more memory-efficient, and cost-effective for cloud-based applications. However, QUILT2 is less robust than QUILT1 for smaller reference panels. We further showed that QUILT2 is about as accurate as GLIMPSE2, being more accurate in certain situations, like lower coverages, and with long or linked read sequencing data. This suggests that QUILT2 should be more accurate than GLIMPSE2 in instances where SNP heterozygosity is high, as would be expected in species with large effective population sizes. In terms of speed, QUILT2 is slower and less efficient than GLIMPSE2 when run at scale; however, it is faster when using very large reference panels to process a small number of samples. As such, QUILT2 offers a benefit over GLIMPSE2 in analysis pipelines geared towards processing samples individually or in small batches, as might happen if imputation is performed immediately after data acquisition.

In addition, we showed with simulated data that QUILT2-nipt can impute both mother and fetus using cfDNA reads from NIPT, which can be very useful in population genetic studies[20,40]. We showed that QUILT2-nipt can achieve high accuracy at common variants in the mother at moderate effective sequencing coverage (e.g. 1×, 0.1 FF). For the fetus, we showed that PRS and GWAS performance are reasonable at moderate effective sequencing coverage (e.g. 1×, 0.1 FF), and that performance increases with sequencing (e.g. 2×, 0.1FF). Given the decrease in sequencing cost in recent years, it is not much more expensive to generate lc-WGS from NIPT at 2× versus the more modest levels common today. Even so, the accuracy of the QUILT2-nipt method at the more modest coverage should allow researchers to leverage existing large NIPT data to conduct GWAS both generally, as well as in specific contexts, such as to help unravel the complicated mechanisms underlying the regulation of human parturition, where both the maternal and fetal genome are involved[41–44]. From a GWAS perspective, the value of maternal and fetal genotypes can be

enhanced through phenotypes acquired through linkage with electronic health records, or through surveys of affected family members, using approaches like GWAS-by-proxy[45] and LT-FH[46]. From a clinical perspective, relevant traits that are important for health care, such as hyperglycemia, gestational diabetes and birth weight, are genetically influenced by both maternal and fetal genomes[14,44,47–51]. Therefore, the imputation of the fetal genome before childbirth affords the opportunity to calculate PRS prenatally, which may impact future health risk management if the PRS is sufficiently predictive, though future studies will be required to fully evaluate this potential. Recent papers, which take the uncertainty of genotype imputation into PRS interpretation into account, are of particular relevance here[52]. Nonetheless, the imputation of specific variants, in particular rare variants, remains challenging. Therefore, it is unlikely to be used for clinical diagnosis, and in that context, recent approaches using non-invasive deep fetal exome sequencing are more promising[22,23]. We note that QUILT2-nipt should still operate on fetal exome sequencing, so in addition to accurate rare variants, such data would also allow for genotype imputation and PRS calculation.

We note that QUILT2 contains limitations and areas for future improvements. First, the QUILT2 cfDNA NIPT results presented here were primarily based on simulated data and were only confirmed using a single real sample. Further work on larger real cfDNA NIPT samples will be necessary to confirm the accuracies and generalizations presented here. Second, it should be possible to model recombination in the maternal haplotypes in NIPT. Although we expect not modeling this to have a small effect for low fetal fractions, and can be mediated by imputing in small genomic windows, future methodological innovations could directly account for this by modeling both maternal haplotypes, and then using a hidden variable reflecting which of the two haplotypes is the maternally inherited one in the fetus. Third, there is clinical value in the detection of de novo CNVs before birth. For example, detection of de novo CNVs at 22q11.2, which occurs in around $1/1500 - 1/2000$ live births[53], allows for delivery to take place at specialized medical centers, and for timely treatment of conditions like for neonatal hypocalcemia and immunodeficiency, which improves outcomes[54,55]. Current methods to detect de novo copy number variation using NIPT focus on either read counting approaches in windows[56], or machine learning approaches using massively multiplexed PCR[57]. By binning reads into sets reflecting their origin, and effectively using long range and external information, it is reasonable to believe an approach based on an extension of QUILT2 could have greater sensitivity and specificity compared to current approaches, and improve clinical utility. Fourth, QUILT2 inherently phases reads to their haplotypic background. We've recently shown that pre-phasing long reads using QUILT2 improves structural variant genotyping[58]. A more complicated, and likely more accurate, caller that modeled a copy number at each genomic location, as would an even more complicated approach that involved iterative re-mapping, would likely improve accuracy further. Finally, as we noted in QUILT1, large lc-WGS datasets offer the possibility to augment reference panels. While reference panels from commonly studied populations, particularly European ones, have large reference panels available, there would be tremendous value in leveraging large NIPT efforts to generate and use novel haplotype reference panels.

In conclusion, we expect QUILT2 to improve imputation of diploid samples in many settings. In addition, the imputation of cfDNA NIPT samples using QUILT2-nipt could enable the creation of large GWAS through recovery of both maternal as well as fetal genotypes, as well as offer healthcare opportunities, enabling prenatal PRS.

## Methods
Our research complies with all relevant ethical regulations.

## QUILT2
Details about the QUILT model have been previously published[4]. In brief, QUILT is a Gibbs sampler that samples read labels based on the observed data (i.e. sequencing reads) and parameters of the model (e.g. haplotype reference panel), and from this, allows for phasing and genotype imputation. Here, we provide information about the mathematical changes to QUILT2 that form the basis of the QUILT2-nipt method. We also give a brief introduction to msPBWT, as well as the iterative process that uses common SNPs then all SNPs. More details about msPBWT, as well as the heuristics used in Gibbs sampling for NIPT are given in the Supplementary Note.

In the NIPT mode, we used Gibbs sampling to generate draws from $H \sim P(H \,|\, O, \lambda, FF)$, where $H$ is a vector of read (haplotype) membership labels, $O$ is our reads (i.e. observed bases at SNPs), $FF$ is the fetal fraction, which is required as an input and can be obtained from NIPT report[59,60], and $\lambda$ is the parameters of our model, including the recombination rate and reference haplotypes. In NIPT, sequencing reads come from three sources, and let us label them as 1 = maternal transmitted, 2 = maternal untransmitted and 3 = paternal transmitted. As such, for each read $v$, we have the prior probability of haplotype membership as

$$P(H_v = h_v | FF) = \begin{cases} 0.5, & h_v = 1 \\ 0.5 - \frac{FF}{2}, & h_v = 2 \\ \frac{FF}{2}, & h_v = 3 \end{cases} \tag{1}$$

Let $o$ be the realized observations for the random variable $O$ (sequencing read bases and base qualities). Consider in the Gibbs sampling that we want to sample a new value for read indexed by $v$ with read label $h_v$, conditional on all other read labels. Let $H_v$ be this random variable at this point in the Gibbs sampler and let $H_{-v}$ be a random variable representing the remaining read labels. We therefore need to calculate:

$$\begin{aligned} &P(H_v = h_v | H_{-v} = h_{-v}, O = o, \lambda, FF) \\ &= \frac{P(O = o, H_v = h_v, H_{-v} = h_{-v} | \lambda, FF)}{\sum_{i=1}^{3} P(O = o, H_v = i, H_{-v} = h_{-v} | \lambda, FF)} \end{aligned} \tag{2}$$

for $h_v \in \{1, 2, 3\}$ and sample $h_v$ using this probability. To do this we use $P(O = o, H = h \,|\, \lambda, FF) = P(O = o \,|\, H = h, \lambda)\, P(H = h \,|\, \lambda, FF)$ which uses the prior probability of all reads, which is the product of the prior for one read described above. This prior probability does not cancel like it did for QUILT1 (diploid) where $P(H = h \,|\, \lambda)$ was constant regardless of $h_v$. For $P(O = o \,|\, H = h, \lambda)$ we note that given the vector of read labels, we split the reads into three sets reflecting their haplotypic origin, call that $O^i$, and can generate probabilities in the normal ways for HMMs, in an obvious extension to what is presented in the QUILT1 paper. For $G_t^i$ the haploid (genotype) for haplotype $i$ for some SNP $t$, we can calculate the posterior probability that this is the alternate allele (encoded by a 1) as follows

$$P(G_t^i = 1 | O, H, \lambda) = \sum_{k=1}^{K} \theta_{t,k} P(Q_{g_t}^i = k | O^i = o^i, \lambda) \tag{3}$$

where, $\theta_{t,k}$ is the probability that reference haplotype $k$ carries the alternate allele at SNP $t$,

$Q_{g_t}^i$ is the reference haplotype carried at grid $g_t$ for haplotype $i$, noting that $P(Q_{g_t}^i = k | O^i = o^i, \lambda)$ is a standard output from an HMM. From these probabilities, a haplotype dosage can be taken as the sum across several Gibbs samplings, and from this, genotype dosages can be taken for the mother by summing up the value for $i = 1$ and $i = 2$, while for the fetus we choose $i = 1$ and $i = 3$.

We have introduced the msPBWT algorithm in QUILT2 to find haplotype matches for target haplotypes against $i$ the full reference

panel, with computational complexity independent of the size of the full reference panel. An overview of msPBWT is given in Supplementary Fig. 1. First, the haplotype reference panel, containing binary information about whether haplotypes carry reference or alternate alleles, is encoded, and then ranked. This ranked matrix contains columns with 255 symbols representing the most commonly seen 32 SNP haplotype in this window (grid), and a final symbol (0) representing a haplotype not amongst the 255 most commonly seen, for which information (the haplotypes carried and their positions) are stored in helper matrices. Next, two indices are built: $A$, which allows for indices to be looked up at any grid to put the original haplotype reference panel in reverse sorted prefix order; and $U$, which allows for determination between grids about how the ordering of indices changes. Both $A$ and $U$ can be used with a two part algorithm that allows for rapid determination of which haplotypes in the haplotype reference panel contain long exact matches to the target haplotype. First, a vector $f$ is found, representing at each grid, where the target haplotype would be inserted into the haplotype reference panel to place the haplotype reference panel in reverse sorted prefix order. Second, scanning up and down is performed, to maintain a list of the closest $L$ haplotypes to the target haplotypes. Finally, filtering is done with respect to a minimum length of match, and when more matches are identified than requested haplotypes, further filtering is done for the haplotypes identified for the two haplotypes, which optimizes both the length of matches, as well as their uniqueness along the target region. Full details of msPBWT are given in the Supplementary Note.

Finally, we note that with high coverage WGS based reference panels, the vast majority of variants are very rare, and thus are not informative early in the iterative process, as they offer little discriminatory information about which reads come from which haplotypes. As such, we introduce a further two-step process, where we first impute using a set of common variants and the reads that intersect them, to get an initial estimate of the carried haplotypes. We then load all reads at all sites, initialize their phase according to the current best estimate of the underlying haplotypes, and perform a final round of read-sampling (Supplementary Fig. 2). Together, these changes made QUILT2 substantially faster and more memory efficient than QUILT1. Finally, we upgraded a heuristic, to perform efficient block Gibbs sampling between all sequential pairs of grids, to identify and solve phase switch errors with read labellings.

## Datasets

We created various test and truth datasets to benchmark all methods. We used three reference panels for testing genotype imputation and phasing accuracy, which are the 200 K WGS derived UK biobank reference panel, the HRC and the 1000 Genomes Project. Additionally, for benchmarking the computational performance, we used the very big reference panel of $N = 976,630$ haplotypes (488,315 individuals), that is the UK biobank data imputed by the Genomics England haplotype reference panel[61], which we referred to as the UKB-GEL. We used liftover to convert the HRC reference panel from the GRCh37 to the GRCh38 build using the Genome Analysis Toolkit[62] (GATK) Picard LiftoverVCF v2.22.2.

*CEU*. We downloaded the high coverage (>30x) CRAM files of 30 CEU trios in the 1000 Genome project, from which we called the true genotypes at sites in the UKB-GEL panel using bcftools[63] v1.18 with option 'call -Aim -C alleles -T ref.sites'. We filtered the true genotypes at sites with read depth lower than 10. Leveraging trios information, phasing was done first assuming Mendelian inheritance, excluding triple-heterozygous sites using bespoke R v.4.2.2 code; then, the excluded sites were phased with this scaffold using shapeit4 v4.2.2[64].

*ONT*. We downloaded high-coverage ONT alignment files of GRCh38 build (https://s3.amazonaws.com/1000g-ont/index.html? prefix=ALIGNMENT_AND_ASSEMBLY_DATA/FIRST_100/) for 11 samples in the EUR super-population from the first 100 ONT samples in the

1000 Genomes Project[65]. When evaluating the imputation accuracy with ONT reads, we used the called genotypes from high coverage short Illumina reads as truth, which was done using the same bcftools command.

*Ancient DNA*. We downloaded four high-coverage ancient DNA samples from the Afanasievo family (mother I3388, father I3950, son I3949, son I6714), who lived ~4.6 thousand years ago (ka) in the Altai Mountains of Russia, with average coverages of 10.8×, 25.8×, 21.2×, and 25.3×, respectively, which have reliable genotype calls and phasing[66]. Since both the public BAM and VCF files are of GRCh37 coordinates, we first liftovered the VCF with true genotypes to GRCh38 and converted the downloaded BAM files back to raw FASTQ using 'samtools fastq' using samtools v1.18. Then we used the mapache[38] snakemake workflow, which implements the best practice for aDNA imputation[7], to map the FASTQ files against the human reference genome GRCh38 to generate the BAM files for imputation benchmarking.

*NIPT cfDNA*. For real NIPT cfDNA data, we found one sample that is publicly available[67], which offers both the sequencing data from cfDNA in maternal plasma and saliva at high coverage (20x and 30x, respectively). We downloaded the raw FASTQ file from https://my.pgp-hms.org/profile/hu058D3E. Following best practice[67], we called the genotypes of the mother using high coverage saliva DNA with GATK, and we generated the BAM file of GRCh38 coordinates from cfDNA of the pregnant women. Additionally, we simulated thousands of NIPT cfDNA with various coverages using the real sequencing reads from trios or duos individuals. The details are described in the next section.

## NIPT cfDNA simulations from real reads

We simulated 3 sets of NIPT cfDNA reads for different analyses using the real sequencing reads. First, to evaluate the QUILT2-nipt and QUILT2-diploid method on maternal genotype imputation, we simulated NIPT cfDNA with various fetal fractions by mixing the maternal reads and offspring reads in the 30 CEU trios from the 1000 Genome Project. We only simulated BAM files for chromosome 20 for benchmarking purposes. Second, to evaluate the GWAS performance on NIPT samples and obtain as many samples as possible, we first identified 1060 trios and 4114 duos in the UK Biobank given the released kinship statistics[37]. Specifically, we defined the first degree relatives as the pairs with kinship value > 0.1767 and IBS0 value < 0.0012. To further identify the trios and duos, we defined that the difference between the parents and offspring in age must be greater than 15. Therefore, given a list of trios and duos, we simulated 11,028 NIPT samples by treating the offspring as the parent regardless of sex and vice versa. We only simulated the BAM files of 11,028 samples for chromosome 1 due to the limited budget and computational resources. Third, to evaluate the PRS performance that requires genotypes of the whole genome, we sampled only 1000 NIPT samples from the above identified duos and trios, which includes 500 unique parent-offspring pairs. For those 1000 NIPT samples, we simulated the BAM files for the whole genome.

## Imputation benchmarking workflow

To access the performance of QUILT2 and GLIMPSE2 (v2.0.0) on various datasets, we developed a reproducible and configurable snakemake workflow for imputation benchmarking (https://github.com/Zilong-Li/lcWGS-imputation-workflow), which includes downsampling BAM/CRAM files to low coverage; creating reference panels by removing the target samples from the input panel; parallelly running by chunks and ligating results; automated reproducible pipeline with support to High-Performance Computing clusters.

## Assessing imputation accuracy

To assess the imputation and phasing accuracy, we developed an efficient function *vcfcomp* in the vcfppR package v4.6.0[68], which can

calculate various concordance metrics between two VCF/BCF files at either samples level or variants level. In the output VCF files, homozygous reference, heterozygous and homozygous alternative genotypes are coded as integer value 0, 1 and 2 respectively, while the genotype dosage are float values between 0 and 2. There are three concordance metrics defined here. First, we defined $r^2$ as the squared Pearson correlation between the imputed dosages (test) and high-coverage genotype calls (truth). Second, we defined the non-reference concordance (NRC) rate as NRC = 1 - (e0 + e1 + e2) / (e0 + e1 + e2 + m1 + m2), where e0, e1, e2 are the counts of mismatches for genotype 0, 1 and 2 respectively, while m1 and m2 are the counts of matches between the imputed and truth for genotype 1 and 2. Third, we defined the F1-score as F1 = 2 * TP / (2 * TP + FP + FN), where TP is the counts of matches between the imputed and truth for genotype 1 and 2, FP is the counts of mismatches for genotype 1 and 2, and FN is the counts of mismatches for genotype 0 and 1.

To assess genotype imputation accuracy at variant level, we first grouped SNP by its frequency intervals and aggregated the concordance metrics (e.g. $r^2$) for all samples across all SNPs in that frequency interval. On the other hand, to assess the genotype imputation accuracy at the sample level, we only aggregated the concordance metrics (e.g. NRC, F1) for genotypes of a single sample at a given allele frequency interval. The allele frequencies were taken from either the gnomAD v.3.1.2 database (for modern European samples, https://gnomad.broadinstitute.org/downloads#v3) or from the reference panel itself (for ancient DNA).

### Assessing phasing accuracy

To assess phasing accuracy, we used phasing switch error (PSE) rates as follows, which is implemented in *vcfcomp* with option *stats=pse*. First, consider that we have both true haplotypes (truth) and the imputed haplotype (test) at sites where the truth genotypes are heterozygous. Next, define as discordant any test sites that are also not heterozygous. On the remaining sites, define a phase switch error when either the true haplotypes record a change where the haplotype carries the alternate allele between adjacent heterozygous sites when the test haplotypes do not or vice versa. We removed from consideration sites that were flipped, that is, yielding consecutive phase switch errors. The PSE rate is the number of phase switch errors divided by the total number of pairs of consecutive heterozygous sites examined and can be combined across discrete imputed windows.

### Association testing and validations

Since the data consists of the related individuals from families, we used the linear mixed model implemented in gemma[69] v0.98.5 for association tests accounting for the genetic relatedness. First, we computed the genetic relatedness matrix (GRM) using *'gemma -gk 1'* based on the genotypes from the array with the default setting for SNP QC (*-maf 0.01 -miss 0.05*). Then we ran association tests on multiple imputed datasets in BIMBAM format (for dosages) using *'gemma -lmm 3'* with the same GRM and sex, age and 10 PCs as covariates. The PCs were calculated based on the imputed array data for the target samples using PCAone[70] v0.4.4. To prepare the BIMBAM format for gemma, we used *bcftools query -f % CHROM:%POS, %REF, %ALT[, %MDS]\n'* to extract the dosages of the mother, and used *bcftools query -f %CHROM:%POS, %REF, %ALT[, %FDS]\n'* to extract the dosages of the fetus from the VCF output of QUILT2.

We used the publicly available summary statistics of GWAS on more than 330 thousand individuals to find a list of the significant association hits in Europeans, which was downloaded from https://www.nealelab.is/uk-biobank (GWAS round 2). Then, to define the independent association signals that we need to validate using our target samples, we used the *extract_instruments* function with default parameters from the TwoSampleMR[71] v.0.5.10 package to perform the clumping and curate a list of independent association signals for each trait.

### PRS and incremental $r^2$

To calculate the PRS, we downloaded the polygenic score of each trait for Europeans[39] from the PGS catalog https://www.pgscatalog.org/publication/PGP000332/. We calculated the PRS for each individual using the imputed genotype dosages and effect size of each trait, which is given by

$$PRS_i = \sum_{j=1} \hat{\beta}_j G_{i,j} \qquad (4)$$

Where $PRS_i$ is the PRS for each individual, $\hat{\beta}_j$ is the effect size associated with each SNP, $G_{i,j}$ is the imputed genotype dosage (float value between 0 and 2) for each SNP and for each individual. Since the array-imputed data used genome builds GRCh37 while the lc-WGS imputed data used GRCh38, we used the harmonized score files that shared the same variants for both genome builds, which resulted in 1,095,616 out 1,109,311 reported SNPs (98.8% matches). We used the nextflow workflow (https://github.com/PGScatalog/pgsc_calc) to automate our analyses[72].

Then, we estimated the prediction accuracy using the incremental R2 (*IR2*), which is the difference in $r^2$ for two regression models for each trait, with one having the PRS and one not having it, with each adjusted for age, sex and 10 PCs.

We ran the regression in R using the *lm* function. Therefore, we can obtain the *IR2* for each trait and each dataset. To estimate the accuracy of PRS using the lc-WGS imputed data compared with the array-imputed data, we defined the relative incremental R2 (*RIR2*) as follows,

$$RIR2 = \frac{IR2_{lc}}{IR2_{array}} \qquad (5)$$

Where $IR2_{lc}$ and $IR2_{array}$ are the *IR2* for each trait calculated with the lc-WGS imputed data and the array-imputed data respectively. Further, to estimate the standard error (*se*) of *RIR2*, we ran the *t.test* in R on a vector of *RIR2* (21 traits).

### Benchmarking and computational cost

To benchmarking the performance of QUILT2 with respect to the size of reference panel, we used the one of the largest reference panel, that is the UK biobank data imputed by the Genomics England haplotype reference panel, which contains N = 976,630 haplotypes (488,315 individuals) and M = 7,179,683 SNPs for chromosome 20. Then we randomly sampled N = 2e4, 2e5, 4e5, 6e5, 8e5 haplotypes to create 6 more smaller panels. To compare QUILT2 against GLIMPSE2, we split the imputation task into multiple chunks using the same chunksize as outputted by the GLIMPSE2_chunk program. Further to fairly measure the average RAM and time for each sample, we ran both software using only one thread, although both support multithreading. To estimate the realistic cost using the WGS-derived UKB-200K panel on the RAP, we maximized the usage of CPUs threads available as possible as we can. The benchmarking was performed on virtual machines of type mem2_ssd1_v2_x4 with 4 cores and 16 GBs memory on the RAP using normal priority. The cost was based on the instance rate card v2 (https://platform.dnanexus.com/resources/UKB_Rate_Card-Current.pdf).

### Reporting summary

Further information on research design is available in the Nature Portfolio Reporting Summary linked to this article.

## Data availability

The 1000 Genomes Project phase 3 dataset sequenced at high coverage by the New York Genome Center is available on the European Nucleotide Archive under accession no. PRJEB31736, the International Genome Sample Resource (IGSR) data portal and the University of Michigan school of public health ftp site (https://share.sph.umich.edu/1000g-high-coverage/freeze9/phased/). The publicly available HRC

reference panel is available from the European Genome-phenome Archive at the European Bioinformatics Institute under accession no. EGAS00001001710. The UKB-200K panel, the UKB-GEL panel, and individuals' WGS data can be accessed via the UKB RAP (https://ukbiobank.dnanexus.com/landing). The ONT alignment files of 11 samples are available at https://s3.amazonaws.com/1000g-ont/index.html?prefix=PROCESSED_DATA. The ancient DNA from Afanasievo culture can be accessed via European Nucleotide Archive, accession number PRJEB43093, and the phased VCF file for the family are available from the European Variation Archive, accession number PRJEB46983. The real NIPT cfDNA data is freely available from the Personal Genome Project: https://my.pgp-hms.org/profile/hu058D3E, and the high coverage sequencing data from saliva of the pregnant mother is available from https://my.pgp-hms.org/profile/huC1F919. The simulated NIPT samples were simulated as described in the Methods section, and code to perform this simulation is available at https://doi.org/10.5281/zenodo.17316024. The GWAS summary statistics for significant association hits in Europeans are downloaded from https://www.nealelab.is/uk-biobank (GWAS round 2), and the GWAS summary statistics in this study based on the simulated NIPT data are available in the Zenodo dataset[73].

## Code availability

QUILT2 is available from https://github.com/rwdavies/QUILT under a General Public License 3 (GPL3). Snakemake workflow for benchmarking lc-WGS imputation is available from https://github.com/Zilong-Li/lcWGS-imputation-workflow. Code to reproduce analysis and figures have been deposited in a Zenodo repository[73].

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

## Acknowledgements

We thank Simon Myers for helpful discussions, and we thank Frederik Filip Stæger for assistance with downloading and preprocessing the GWAS summary statistics. We thank all participants in the UK Biobank. Z.L. and A.A. were supported by the Novo Nordisk Foundation (NNF20OC0061343). This research has been conducted using the UK Biobank Resource under Application No. 32683.

## Author contributions

Z.L. and R.W.D developed and implemented the method as well as conceiving the study. Z.L. and R.W.D. performed the analyses. A.A. contributed to simulation, GWAS, and PRS analyses for NIPT data. Z.L., A.A., and R.W.D. wrote the paper. All authors reviewed and approved the final manuscript.

## Competing interests

R.W.D. became a full time employee of Genomics Ltd during the drafting of this manuscript. The other authors declare no competing interests.
