## [Transparent Peer Review file · Nature Communications]

Flexible read-aware genotype imputation from sequence using biobank sized reference panels

Corresponding Author: Dr Robert Davies

Version 0:

Reviewer comments:

Reviewer #1

(Remarks to the Author)

The manuscript presents QUILT2, a genomic imputation tool, demonstrating enhanced scalability, computational efficiency, and accuracy over existing methods. It highlights QUILT2's performance across diverse scenarios, including short and long sequencing reads, ancient DNA, and non-invasive prenatal testing (NIPT), where it imputes both maternal and fetal genotypes.

I really appreciate the quality of the work; it's very well presented in the paper, and the accompanying online resources (like the Snakemake workflow (!), tutorials, and comprehensive documentation) are excellent. Thanks!

Major Comments:

The authors mention two technical advancements (reduced computational complexity and two-stage imputation). How do these changes to the Positional Burrows-Wheeler Transform (-> resulting in msPBWT) influence the trade-off between speed and accuracy compared to PBWT? Also, what exactly contributes to QUILT2's improved accuracy, considering that GLIMPSE also uses PBWT (to the best of my knowledge, not stated in the manuscript)?

Is QUILT2 intended to fully replace QUILT1 for all reference panels / use cases, or are there cases where QUILT1 remains preferable? Maybe this should be stated in the manuscript.

Why is the imputation of long reads generally less accurate, as shown in Figure 2? Is this somewhere mentioned?

The authors claim that QUILT2 significantly outperforms GLIMPSE2 in scenarios where a single sample is imputed with a large reference panel. Can they elaborate on what makes QUILT2 so efficient in this context?

When using QUILT2 on the UK Biobank Research Analysis Platform (RAP), are additional pre-processing steps needed for the reference panel? (e.g. QUILT2_prepare_reference.R) Specifically, does the panel need to be in VCF format, and if so, is this format already available on RAP (I guess not)? Are there post-processing steps for QUILT available (e.g. ligate)? I think users would highly appreciate a tutorial how to create reference panels (e.g. locally or on UKB) run the imputation process + post-processing steps. How does the chunk size affect the quality? This could be interesting when conducting a WGS imputation for example on a local Slurm cluster.

For the new QUILT2-nipt2 approach, is the fetal fraction required as an input parameter? Is this somewhere stated?

Ancient DNA is briefly mentioned in the abstract but lacks focus in the introduction or in the title. Could the authors provide more context earlier in the manuscript to align the emphasis on ancient DNA with its later discussion? Or to state it differently, is the use case of ancient DNA central to the manuscript?

Given the reliance on simulated NIPT data, how closely do these simulations mirror real-world scenarios? Are there potential biases that might affect the conclusions?

Are there specific genetic architectures or data types where QUILT2 might be less effective compared to other imputation

methods?

The results suggest that QUILT2-nipt could significantly enhance GWAS and PRS performance using imputed fetal genotypes. Could the authors elaborate on the implications for clinical applications, particularly the accuracy requirements in prenatal risk assessments? Are there recommended r^2 thresholds for polygenic score calculations based on fetal fraction? Given the ongoing debate around the clinical utility of PGS and the uncertainty of fetal genotypes (depending on seq depth + FF, see also below), this topic could result in diverse opinions.

Minor:

The abstract describes the fetal genome imputation results using QUILT2 at 0.25× sequencing coverage with a 10% fetal fraction as “accurate,” yet the r^2 is only 0.465. This statement seems potentially misleading—could the authors clarify this assessment?

Line 154: Please confirm the minor allele frequency (MAF) levels used to define common SNPs.

There are several typos in the manuscript that need to be fixed.

(Remarks on code availability)

See above. I think the quality of the Github repo is very very high.

Reviewer #2

(Remarks to the Author)

Summary of the key results, Originality & significance

Li et al. reported an improved method for genotype imputation in data produce from short- and long-read sequencing data, named QUILT2. The methodology is primarily based on their previous work (QUILT) that makes use of sequencing reference data to impute undetermined genotypes of shallow-sequencing data (PMID: 34083788). The authors stated two technical and a methodological improvement over their previously published works (PMIDs: 27376236, 34083788). Technical improvements include: (i) a memory efficient PBWT with the purpose of finding haplotypes in the reference panel that shares long matches with constant computational complexity and “very low” memory footprint; (ii) a two-stage imputation approach that first samples read labels and makes use of common SNPs to find an optimal subset of haplotype reference panel. Methodological improvement includes assumes that three haplotypes are present in the NIPT data.

This reviewer did not find these improvements conceptually novel relevant to their previous works (PMIDs: 27376236, 34083788) and the existing work GMPSE2 (PMID: 37386250). In additions, although the authors claim applicability of their method (three-haplotype option) for non-invasive prenatal testing (NIPT) by discerning fetal and maternal genomes and imputing the fetal genome, in my view, the assumptions are far from real cfDNA component in blood stream of pregnant women (see below). And that it has been performed on simulated data from whole-genome sequencing data of trios. For the latter, the r^2 values are specifically too low to claim that the methodology works on low-pass sequenced NIPT data.

Major comments:

- The QUILT2-nipt method assumes the observed sequencing reads are distributed among three haplotypes (maternal transmitted, maternal untransmitted, and paternal transmitted). However, the complexity of real-world data, including recombination in maternal haplotypes and errors in fetal fraction estimation, may introduce inaccuracies that are not addressed. How do you address recombination sites between the maternal homologs?

- Given my comment above, it is incorrect to assume that NIP data has “triploid nature”, it is rather tetraploid (prenatal chromosomes recombine).

- R^2 is very low for real NIPT data. Therefore application of QUILT2 can lead to wrong SNP calls. Could you clarify which parts of genomes cause this low r^2 value?

- The method has higher accuracy for common variants but its applications with (very) rare variants is challenging and substandard. While this is expected, the reliance on filtering out rare variants early in the iterative imputation process could overlook clinically relevant but less frequent variations. Specially in the context of NIPD. Please clarify how you would deal with this challenge?

- The authors discuss the potential clinical utility of imputed fetal genomes for traits such as gestational hypertension and preeclampsia but does not provide sufficient empirical evidence or validation studies to support these claims. Without these and my concerns above, this reviewer finds such phrases more over-statements.

- The authors briefly state the potential of QUILT2 to identify structural variations and copy number variants but did not provide how or a benchmarking for these applications. Please clarify.

- For the diploid imputation performance, the authors indicate that they “used Chr 20 for convinience”. Why did you make such choice? Is this then correct that all the data presented in this manuscript are using Chr 20 only?

- The authors devised a 32-bit grid strategy and allowed transition after 32nd pair of SNPs. It is not clear to this reviewer why 32 and how the transitions have happened?

- Description of QUILT2 for (ONT) long-read sequencing sounds preliminary. Please clarify how you make use of long-read sequencing data and kb-Mb size haplotypes. How it differs from short-read sequencing? How it performs for structural rearrangements? How do you potentially detect erroneous SNVs of the ONT data?

Minor comments:

- In the beginning of the Results section "...QUILT2 is based upon, and improves, our previous model QUILT1 for diploid genotype imputation, which operates, which uses an iterative approach...". Either use "which operates" or "which uses"

- Please make figures that illustrate QUILT2, clearer. Specially for the grid search and using common SNPs as anchors.

- The authors state that "In some countries, NIPT is routinely offered to all pregnant women, for instance in the Netherlands, with an average uptake rate of 46%". This is not up to date. This uptake rate is from the TRIDENT study that was finalized in March 2023. Since April NIPT in the Netherlands is a screening program with very high uptake rate. Please correct.

- Please insert page and line numbers to make the review process easier and more efficient.

(Remarks on code availability)

There are different GitHub repositories referred in "Code availability" section of the manuscript. Please make a clear scheme how they are related to each other.

The codes require extensive documentations, indicating what function or piece of code does what. For instance, "## known ped, weird format" is not really informative.

Reviewer #3

(Remarks to the Author)

The article presents QUILT2, a method for genotype imputation from low-coverage whole-genome sequencing (lc-WGS), building on the QUILT1 framework. QUILT2 integrates a memory-efficient implementation of the positional Burrows-Wheeler transform (msPBWT) and a two-stage imputation process, aiming to improve the speed and scalability of imputation. Additionally, it introduces a novel approach for imputing both maternal and fetal genomes from non-invasive prenatal testing (NIPT) data. While the method shows promise in some applications, especially for NIPT data, the manuscript raises several concerns.

Biased benchmark

Overall, the benchmark puts emphasis on QUILT2 outperforming GLIMPSE2, while it seems to avoid showing scenarios where GLIMPSE2 may be preferred. For example, it is unclear if GLIMPSE2 demonstrates better performance at higher coverage levels, as the benchmarking ends at 2x coverage, where differences seem to favor GLIMPSE2. This omission limits a comprehensive understanding of QUILT2's strengths and weaknesses. Additionally, computational time is reported at 0.25x coverage, where GLIMPSE2 is shown to be about four times faster, but it is unclear if this difference is maintained at higher coverage. A more comprehensive and fair evaluation of methods across coverage levels and contexts would enhance the manuscript's scientific validity.

Lack of citations to previous work

The authors seem to overstate the novelty of their approach by not clearly citing relevant prior work underlying QUILT2. Several key computational features of the QUILT2 method have been previously introduced by other methods, yet there is a noticeable lack of acknowledgment of these contributions.

The msPBWT shares a lot of similarities with the multi-allelic PBWT (Naseri et al, 2019) and/or Syllable-PBWT (Wang et al, 2023).

PBWT for haplotype selection has been used in numerous methods already: EAGLE2, SHAPEIT4, SHAPEIT5, IMPUTE5, GLIMPSE1, and GLIMPSE2.

Treating rare and common variants differently to speed up computation has been used in BEAGLE5, IMPUTE5, SHAPEIT5, and GLIMPSE2.

Also, not discussing methodological differences between GLIMPSE2 and QUILT2 is a significant oversight given the similarities between the two methods. The first mention of GLIMPSE2 occurs in the method comparison and does not provide any context for this comparison.

Presentation of results

I believe the whole Figures 2 and 3 should be carefully rethought and redesigned for clarity and more importantly for transparency.

- Figure 2a: The comparison of methods for coverage > 0.25x is simply not visible.
 - Figure 2c: The choice of 0.25x coverage is unclear. It seems that the difference between methods do not really differ to what is already shown in 2a. What does the comparison look like for 1x coverage?
 - Figure 2d: Accuracy is presented differently for this analysis compared to 2a-c. That makes results harder to read.
 - Figure 3a-b: These two figures need to show computing time for datasets comprising thousands of samples and even more, cases where compute time and memory usage are a real issue.
 - Figure 3c: Why #samples=8 here and why stop at 1x while most plots show results up to 2x?
- Finally, it seems unfair to choose the reference panels for which QUILT2 performed the best in some panels of fig. 2: 2a shows results for UKB, while 2d for 1000G. The authors should either show results for all reference panels as the main figure or pick one and stick to it.

Lack of reference for major common use cases

Major low-coverage WGS initiatives, like Blended Genome-Exome (BGE), focus on higher sequencing depths (2-4x), yet the manuscript does not address QUILT2's application at these depths for large-scale cohorts. I think all benchmarks should at least include 4x coverage configuration beside the coverages already presented.

The timing benchmark seems also to miss the point: compute time for single-sample imputation is not anymore an issue. Scalability for hundreds of thousands of samples as in typical GWAS is actually the critical aspect. This is not addressed in the benchmark, nor in the discussion. Readers will want to know how running times and RAM usage scale on large datasets comprising many thousands of samples.

Ancient DNA benchmark

The authors do not describe how imputation has been run on ancient DNA. Established pipelines involving GLIMPSE are already used in real-world ancient DNA applications, with GLIMPSE2 authors explicitly advising against its simplistic genotype likelihood model for complex scenarios like aDNA. If the authors choose to retain the ancient DNA benchmark, they should compare their tool with the recommended pipeline outlined by Mota et al., Nat Comm. 2023.

Long reads

The application of QUILT to long reads is a key advantage, but this was demonstrated in prior work. It is unclear why similar results are highlighted in a main figure here or what this benchmark aims to achieve. If the focus is on long-read performance, the comparison should emphasize improvements over QUILT1, rather than contrasting with GLIMPSE2, which is not designed for long reads.

NIPT analysis

We find this part of the manuscript useful to the community, nicely carried over and interesting. This is a significant advance in the field.

Minor comments

- The imputation performance of QUILT2 at indel sites is not clear. Could the accuracy for SNPs and indels be separated?
- The statement "For both human and animal studies, QUILT has been shown to be the most accurate and robust approach when a reference panel is available" is overly assertive. Both the GLIMPSE2 study and this manuscript suggest minimal performance differences between methods.
- The claim "QUILT2 should retain greater accuracy over GLIMPSE2 in various non-human settings, as the effective population size, and hence SNP density, is often much higher than in humans" requires further clarification. Where is the data showing this?
- The statement "We also showed the extent to which large haplotype reference panels can improve both phasing and imputation performance" does not represent a major contribution of this work. This has been demonstrated previously when the reference panel was introduced (citation missing), and this study largely reaffirms those findings.
- References for some reference panels (HRC, UKB, etc ...) are missing.

(Remarks on code availability)

The code is both accessible and functional. Additionally, the documentation is comprehensive and thorough.

Reviewer #4

(Remarks to the Author)

The authors introduce QUILT2, a method that builds on the original QUILT1 for accurate genotype imputation using reference panels. QUILT1 has a linear computational complexity that scales with reference panel sizes and number of

SNPs, both of which are becoming increasingly large, thereby slowing the method runtime. QUILT2 addresses this with a scalable, memory-efficient design for low-coverage WGS imputation. Notably, it also enables imputation of fetal versus maternal genomes from cell-free DNA in non-invasive prenatal tests—an underexplored application. The paper effectively demonstrates the method's robustness and efficiency, though a few comments are provided below.

Major comments:

1. Regarding framing, NIPT is a useful application in its own right, but the framing that it will substantially improve the ability of GWAS to reach millions of participants is somewhat of a funny one, particularly as most phenotypes studied by GWAS are adult-onset diseases or adult quantitative traits. Maybe reconsider NIPT framing as contributing substantially to GWAS in the introduction? Relatedly in the discussion, it is not clear how having common variants from a fetus would change clinical practice in cases of maternal preeclampsia or gestational hypertension, so perhaps clarifying or removing would be helpful.

2. Relatedly, is using fetal genomes in a GWAS / PRS feasible at current accuracy rates? Variants at 1% MAF are only a little over 60% accurate even at 4x coverage and 0.3 FF. Could it introduce false positives? Most of the significant hits are only nominally significant both for mother and fetus (Fig 5b). Similarly, with the current state of PRS as predictors for traits, does matching array data performance with cfDNA really change the feasibility of its use in the clinic? This is an interesting and valuable analysis, but some text in the discussion addressing this caveat could be merited. Could include the study “Novel insights into the genetic architecture of pregnancy glycemic traits from 14,744 Chinese maternities” by Z. Zhu et. al. (2024, Cell Genomics) in the discussion relating to using NIPT data in GWAS.

Minor comments:

- The manuscript could benefit from a careful read through to streamline and improve readability as there are some very complex sentences that are hard to parse. Some examples:
 - The first sentence of results lacks clarity: QUILT2 is based upon, and improves, our previous model QUILT1 for diploid genotype imputation, which operates, which uses an iterative approach, as follows. Suggest to remove “which operates” from the sentence.
 - There are two i.e.'s in this 55-word sentence. It would be more readable if broken up or streamlined: “However, state-of-the-art imputation methods for lc-WGS are designed for diploid samples, i.e. where the sequencing reads come from two haplotypes 3–6, and there is no dedicated method designed for utilizing the triploid nature of NIPT data, i.e. where the sequencing reads come unequally from the maternal transmitted, maternal untransmitted, and paternal transmitted haplotypes.”
 - “For diploid samples, we used the WGS derived UKB panel to show that QUILT2 is approximately as accurate as QUILT1, but substantially faster, and uses much less memory, and as such, should be much less expensive in cloud based applications.” Example streamlined suggestion: For diploid samples, the WGS-derived UKB panel showed that QUILT2 is as accurate as QUILT1 but significantly faster, more memory-efficient, and cost-effective for cloud-based applications.
 - “In addition, while QUILT2 is slower than GLIMPSE2 when processing many samples, it is faster, by an order of magnitude on very large reference panels, when processing only a very small number of samples.” Suggestion: While QUILT2 is slower than GLIMPSE2 for many samples, it is an order of magnitude faster when using very large reference panels to process a small number of samples.
 - Line 99-101: suggest that the sentences “Because of the second part of the iterative process, QUILT1 has linear time complexity in the number of haplotypes and variants in the full reference panel. For RAM and speed reasons...” be rewritten to “Given the two-part iterative process, QUILT1 has linear time complexity dependent on the number of haplotypes and variants in the full reference panel. To minimize RAM and runtime...”
- Including the number of SNPs as another factor in the runtime / memory usage could be interesting. For example, subset the SNPs in the lc-WGS data and see how it varies with the number of SNPs included. There is some analysis on this for the QUILT-nipt method vs QUILT2, though it only focuses on all variants vs only common, and does not mention the number of SNPs included at these thresholds.
- Given that low-coverage WGS imputation methods often differ from GWAS array imputation methods in that the former relies on genotype probabilities whereas the latter relies on genotype calls, is QUILT2 incompatible with GWAS array data? This point is worth clarifying to ensure use cases are clear to readers.
- Common variant definition of $MAF=0.1-0.2$ seems a little funny. Why not just $MAF > 0.1$?
- Given that QUILT2 imputed more accurately than GLIMPSE2 with UKB WGS but not phase 3 1kGP and HRC, is there better genotype error tolerance in GLIMPSE2? Notably, a 30x WGS version of 1kGP is available.
- In Figure 2a, it is hard to tell the coverage lines apart because points overlap. The authors might consider using a color gradient for increasing coverage, with QUILT2 vs GLIMPSE instead indicated by line type (e.g. dashed vs solid). Comparing accuracy across Figure 2a, 2b, and 2c is challenging as the scale is changing.
- Very interesting to see the ancient DNA imputation results. Though in Figure 2d (and corresponding supplemental figures) - is there a particular reason to switch from MAF as x-axis (for figures 2a-c) to coverage as x-axis and color as MAF in Fig2d? Could consider switching the axis to match the other plots, though not critical to do this.

8. Non-reference concordance is a bit more of a forgiving metric than aggregate r^2 . Do the authors care to comment on the relevance of this in the comparison of aDNA vs modern samples?

9. In Figure 3, the legends are a bit confusing, since those shown in A differ from in B but are relevant to A-C. I would suggest extracting a common legend to the bottom of this figure, then reducing the y-axis for panel b, as it is hard to distinguish between lines. The description of memory usage in the text describing this figure could be clearer about the relative comparison between QUILT2 and GLIMPSE2 (note that GLIMPSE2 is not mentioned here): "We note that QUILT2 uses slightly higher RAM given 100 samples to run, and QUILT2 also runs faster with lower RAM for lower coverage data (Figure 3C, Supplementary Figure 5, 6)."

10. The NIPT comparisons in Figure 4 are nice. The authors might want to comment on whether imputing only common fetal variants is the best strategy given the focus on embryonic screening for structural variants and potentially Mendelian variants, or whether lcWGS is an uncommon current application but with approaches like QUILT2 enabling more.

11. Any thoughts on why there is a downward trend in aggregated R^2 for common SNPs in Fig 4, both maternal and fetal genomes? More pronounced for diploid than NIPT and more pronounced with increasing coverage.

12. Why were only quantitative traits used for the GWAS / PRS analyses? Do the authors expect a change in behavior for logistic mixed models? Don't necessarily have to test binary traits, but just include a sentence explaining why quantitative should behave the same as binary traits.

13. For the NIPT GWAS locus identification part, the authors use a loose definition of $p < 0.05$ to describe identification. Figure 5B shows that the vast majority are identified with nominal significance, so a concordance check on effect size (same direction at least?) may also be warranted. It might be helpful to reorder these bars so that the mother array followed by mother with decreasing coverage are shown, followed by the simulated fetus, as it's otherwise hard to compare the relative change in performance. Blue and orange coloring in Figure 5B should be switched to not be confused with the blue/orange for fetus/mother color scheme in other plots. In terms of relative gains in discoverability expected across technologies, this publication might be worth discussing: Gaynor, S.M., Joseph, T., Bai, X., Zou, Y., Boutkov, B., Maxwell, E.K., Delaneau, O., Hofmeister, R.J., Krasheninina, O., Balasubramanian, S., et al. (2024). Yield of genetic association signals from genomes, exomes and imputation in the UK Biobank. *Nat. Genet.* 56, 2345–2351.

14. Given the focus on NIPT, how many common clinically actionable HCMG genes or other common variants relevant to fetal Mendelian disease (e.g. CFTR deltaPhe508) are well-imputed? Is low-coverage WGS the best strategy here, or might a WEGS or blended strategy that enriches for higher coverage at interpretable coding variants be worth discussing?

15. The discussion point about the clinical actionability of de novo CNVs e.g. 22q11.2 is interesting but also confusing, as imputation methods like QUILT2 use a reference panel to only impute existing variants in the population. This is framed as a potential future development opportunity for QUILT2 that seems highly speculative at present as a class of variation undetectable by any imputation method.

16. Methods: " λ is the parameters of our model" - this parameter needs to be defined more clearly.

17. Line 623: "Real big" reference panel - change to "very large" or "one of the largest"

18. Not sure if it's the pdf formatting, but for the notation in the supplemental note, there is no semi-colon.

(Remarks on code availability)

The README provides detailed instructions, including conda environment installation, test run information, and tutorial pages for the various use cases of QUILT2. A number of issues have been opened on the github, which is normal with any bioinformatics software. Importantly, the authors seem responsive to those ongoing issues. We did not try to install the software, but based on the github repository, the usability seems high.

Reviewer #5

(Remarks to the Author)

(Remarks on code availability)

Version 1:

Reviewer comments:

Reviewer #1

(Remarks to the Author)

The authors have provided clear and thoughtful responses to my comments (Reviewer #1), addressing both major and minor concerns. There are no further comments from my side.

Nevertheless, I was not fully convinced by their reply to Reviewer 3's comment on "Lack of reference for major common use cases," specifically regarding the timing benchmark. Please find below my reply to this point:

REPLY TO ANSWER "Lack of reference for major command use cases" regarding the Timing benchmark:

I would like to add a further comment regarding large-scale GWAS analysis. I understand the rationale for splitting jobs across samples, which provides a runtime advantage for QUILT2 when dealing with small sample sizes. In your response to Reviewer 3 regarding large scale GWAS analysis, you note: "As such, the issue of running large datasets with many thousands of samples is really just an issue of how to most efficiently run per-sample, which is why we've focused as we have on comparatively small sample sizes like shown in Figure 3."

If I understand correctly, the authors suggest splitting large datasets both into chunks and by sample. My concern is that this approach introduces substantial overhead in starting and managing multiple machines, as it generates a large number of tasks per imputation job, potentially offsetting any runtime advantage. While fast per-sample processing can be beneficial, higher parallelisation does not make it automatically faster in a cloud environment (though it could be advantageous on a local Slurm cluster). I am curious whether the authors could provide an overall estimate of cloud-based imputation performance (e.g., using a larger sample size for both methods -> runtime, costs). By comparison, current state-of-the-art imputation servers using microarray data typically use chunking only, which works efficiently in the cloud. Even when using spot instances, which may be interrupted more frequently, the cost advantage is substantial (I think up to 90% cheaper than on-demand instances). I would also appreciate clarification on whether the Snakemake pipeline supports per-sample chunking, as this is not as straightforward for end-users as region-based chunking (due to splitting and merging in correct order). Finally, I am unfamiliar with the distinction between "high" and "low" priority jobs. In the UK Biobank instance rating card, the main difference is between spot and on-demand instances (0.2336 vs 1.2832 GBP/hour for the instance type used), not sure if these terms can be used interchangeably.

(Remarks on code availability)

Reviewer #2

(Remarks to the Author)

Thanks to the authors for addressing my comments.

Given their reply and that they only showed that their approach partly may work in an NIPT context (i.e. only one sample). If the editor decides to publish this manuscript, I recommend to remove NIPT (fetal cell-free DNA) in the abstract and the title and instead limit their claims as potential application of QUILT2 in an NIPT context. The low imputation capability and testing QUILT2 on only one actual NIPT sample, is far from reality.

In my opinion, QUILT2 is far from "enabling imputation of the maternal and fetal genome using cell free non-invasive prenatal testing (NIPT) data."

(Remarks on code availability)

Reviewer #4

(Remarks to the Author)

The authors have been responsive to my (and seemingly other) reviewer comments, and I support the advances of QUILT2 moving towards publication.

(Remarks on code availability)

The code and documentation were already quite readable, and the authors appear to continue being responsive to issues as they are opened.

Reviewer #5

(Remarks to the Author)

(Remarks on code availability)

Reviewer #1 (Remarks to the Author):

The manuscript presents QUILT2, a genomic imputation tool, demonstrating enhanced scalability, computational efficiency, and accuracy over existing methods. It highlights QUILT2's performance across diverse scenarios, including short and long sequencing reads, ancient DNA, and non-invasive prenatal testing (NIPT), where it imputes both maternal and fetal genotypes.

I really appreciate the quality of the work; it's very well presented in the paper, and the accompanying online resources (like the Snakemake workflow (!), tutorials, and comprehensive documentation) are excellent. Thanks!

Thank you for taking the time to review our manuscript and for the positive comments.

Major Comments:

The authors mention two technical advancements (reduced computational complexity and two-stage imputation). How do these changes to the Positional Burrows-Wheeler Transform (-> resulting in msPBWT) influence the trade-off between speed and accuracy compared to PBWT? Also, what exactly contributes to QUILT2's improved accuracy, considering that GLIMPSE also uses PBWT (to the best of my knowledge, not stated in the manuscript)?

PBWT has constant computational complexity in haplotype reference panel size, and so does msPBWT, so in that sense, there is no trade off in speed. However, msPBWT is better in that it is much more memory efficient, and should be slightly faster, compared to PBWT, at the slight expense of accuracy. The changes are motivated as msPBWT is a memory-efficient variant of PBWT that is designed to work around one of the key assumptions that facilitates rapid imputation in QUILT and QUILT2, where recombinations are disabled between except between every 32nd SNP. For densely imputed data, where recombinations occur infrequently compared to the number of SNPs, there should be nearly no loss in accuracy in using msPBWT vs PBWT, just gains in RAM and speed.

More generally, gains in accuracy in QUILT2 vs GLIMPSE2 are due to the underlying read-aware model in QUILT2 vs per-SNP independent genotype likelihood in GLIMPSE2, as well as potentially how the msPBWT / PBWT results are used by QUILT2 and GLIMPSE2 to select the new set of conditioning haplotypes. Gains in accuracy in QUILT2 (using msPBWT) vs QUILT1 (using a Li and Stephens model) are due to the much larger haplotype reference panels that QUILT2 is able to use.

We've made it more clearer now in the Introduction and Discussion by presenting the methodological differences between the methods.

Is QUILT2 intended to fully replace QUILT1 for all reference panels / use cases, or are there cases where QUILT1 remains preferable? Maybe this should be stated in the manuscript.

QUILT2 is not intended to fully replace QUILT1 as any msPBWT / PBWT based approach will be faster but less robust. As an example, consider random very rare genotyping errors in the haplotype reference panel. Using a Li and Stephens model, these will contribute very little to the copying probabilities when selecting new conditioning haplotypes, so can be ignored. However, PBWT style

approaches require exact matches, so any single genotype error will break long exact haplotypes, hampering the ability of the model to use them in the next round of imputation.

As such, QUILT1 would be preferred in situations where there are smaller or less accurate haplotype reference panels, for instance in non-human studies. We now made this result more clearer in the text.

In the second paragraph of "Diploid imputation performance", we highlighted: For imputation of modern samples with smaller panels (KGP and HRC), we found that QUILT2 is less robust and accurate than QUILT1 and GLIMPSE2 for rare variants with default settings (Supplementary Figure 1).

In the Discussion, we noted: For diploid samples, the WGS-derived UKB panel showed that QUILT2 is as accurate as QUILT1 but significantly faster, more memory-efficient, and cost-effective for cloud-based applications. However, QUILT2 is less robust than QUILT1 for smaller reference panels.

Why is the imputation of long reads generally less accurate, as shown in Figure 2? Is this somewhere mentioned?

We suspect the decrease in accuracy for the long read analysis shown in Figure 2 is predominantly driven by the higher error rate of the sequenced bases in the reads and the error in long read alignment. Secondly, at low coverage, longer reads are at a disadvantage in terms of coverage of the two diploid haplotypes in normal imputation. As an extreme example, if the average read length is 100 kbp, then at 0.1X, it would be fairly common to have 0 sequencing reads from one haplotype over a 1 Mbp span.

We've previously mentioned this in the QUILT1 paper, saying *This explains why short-read (Illumina-based) sequencing outperforms long-read data (exemplified by ONT) because the slightly improved read label assignment possible from long reads is outweighed by the much lower per-base error rate of the short-read data.*

We think it's probably best not to mention this again in the text in this manuscript but are willing to do so if the reviewers think it wise

The authors claim that QUILT2 significantly outperforms GLIMPSE2 in scenarios where a single sample is imputed with a large reference panel. Can they elaborate on what makes QUILT2 so efficient in this context?

To be clear, this is in terms of speed not accuracy, as both methods impute samples independently from each other. As for speed, to the best of our understanding of how GLIMPSE2 works, this is caused by how both methods read in and process the reference haplotype data.

With QUILT2, haplotype reference panel processing is done beforehand, including making the necessary data structures for msPBWT, and the resulting data structures are stored as simple binary objects. Loading these up afterwards for any subsequent run is very fast and does not contribute meaningfully to run time.

However, the process by which GLIMPSE2 loads and makes available the reference data for later PBWT based analyses is relatively slow and has linear computational complexity in the number of haplotypes

in the haplotype reference panel. This fixed time cost contributes only minimally with large sample size, but contributes substantially at small sample sizes.

Hence why at low numbers of samples, QUILT2 can be substantially faster (more efficient) than GLIMPSE2.

When using QUILT2 on the UK Biobank Research Analysis Platform (RAP), are additional pre-processing steps needed for the reference panel? (e.g. QUILT2_prepare_reference.R) Specifically, does the panel need to be in VCF format, and if so, is this format already available on RAP (I guess not)? Are there post-processing steps for QUILT available (e.g. ligate)? I think users would highly appreciate a tutorial on how to create reference panels (e.g. locally or on UKB) run the imputation process + post-processing steps. How does the chunk size affect the quality? This could be interesting when conducting a WGS imputation for example on a local Slurm cluster.

Thank you for the comments. On the UKB RAP, there are reference VCF files available, which are generated and phased by BEAGLE5 and Shapeit5. The snakemake workflow we developed has the flexibility and the capability for pre-processing the reference panel (e.g. split the genome into chunks) and post-processing the results (ligating the chunks, measuring the accuracy and generating reports if a truth VCF is specified). So while we have not provided an explicit tutorial, the snakemake workflow should help users operate in the UKB RAP environment. Dedicated tutorials can be found elsewhere (e.g. from BEAGLE5). Note that in the benchmarking, we didn't include the cost of additional pre-processing or post-processing steps if there are any, for any of the methods.

In terms of the chunk size, most imputation methods should be robust to the chunk size being too small, as long as the buffer is sufficient. Similarly, since most methods use approaches with conditioning haplotypes, the region should not be too large. In practice, windows of about 2-5 Mb seem appropriate for most modern methods, including QUILT2.

For the new QUILT2-nipt approach, is the fetal fraction required as an input parameter? Is this somewhere stated?

Yes, fetal fraction is required as an input parameter, and yes, this is stated on the website under the NIPT section, where it's listed as a requirement "fflist" which is a file with a one-to-one correspondence with "bamlist" or "cramlist". We've now added an explicit statement in the second paragraph in Methods "*which is required as an input and can be obtained from NIPT report^{52,53}*", which includes references to how to calculate it including the paper "Bioinformatics Approaches for Fetal DNA Fraction Estimation in Noninvasive Prenatal Testing." We chose to use an external value in QUILT2, rather than re-estimate it, for simplicity, and as it is easy to calculate using existing tools. We also showed QUILT2 is robust to misspecification of this parameter in the text.

Ancient DNA is briefly mentioned in the abstract but lacks focus in the introduction or in the title. Could the authors provide more context earlier in the manuscript to align the emphasis on ancient DNA with its later discussion? Or to state it differently, is the use case of ancient DNA central to the manuscript?

Thank you for the comments. We think showcasing the imputation performance of QUILT2 on ancient DNA advances a central theme of this manuscript that QUILT2 is able to accurately handle many different types of sequencing input and applications. In addition, the imputation of ancient DNA can also benefit from large reference panels, which was shown by the GLIMPSE2 paper and was also demonstrated by our results. We've added more text to this motivation in the Introduction

Additionally, imputation with reference panels has been demonstrated as a reliable approach for ancient DNA (aDNA) studies 7. Leveraging large reference panels, aDNA imputation can be further improved 6. Furthermore, human genetic studies have begun using long-read sequencing more and more for investigating disease-associated variants, and strategy of mixing high and low coverages is favored for population-scale study 8.

Given the reliance on simulated NIPT data, how closely do these simulations mirror real-world scenarios? Are there potential biases that might affect the conclusions?

Thank you for the comments. We chose this kind of simulation using real trio or duo haplotypes because the simulated NIPT data should then have the exact same recombination patterns between the fetus and the mothers genomes that are present in NIPT data. This means that we do not have to make assumptions about recombinations in meiosis, and so in that regard, we're mirroring very closely the real-world case. However, at the molecular level, there is a difference between the cfDNA in the plasma of pregnant women and saliva DNA of non-pregnant adults (the one we used for simulation). It has been observed that fetal-derived cfDNA fragments tend to be shorter and enriched at different locations than the maternal cfDNA fragments (<https://pmc.ncbi.nlm.nih.gov/articles/PMC7935715/>). In this revision, we analyzed one real whole genome sequenced NIPT sample, which has 20X coverages of plasma cfDNA with estimated 6% FF and 30X coverages saliva DNA (Extended Figure 4), which confirms the effectiveness and improvements of QUILT2-nipt over QUILT2-diploid for imputing the mother.

*In the first paragraph of "NIPT cfDNA imputation performance", we added:
In addition to simulations, we also confirmed the effectiveness and improvements of QUILT2-nipt over QUILT2-diploid for maternal genotype imputation with one real NIPT sample (Extended Figure 4).*

Are there specific genetic architectures or data types where QUILT2 might be less effective compared to other imputation methods?

Thank you for the comments. Most imputation methods can either only work on a single data type, or are highly optimized for that data type. For instance QUILT2 and GLIMPSE2 for sequence based imputation, while other programs like Beagle (recent versions), Shapeit and IMPUTE are designed for genotyping microarray based imputation. As such, in designing a study, a researcher would first choose the genotyping method (sequence or arrays), and then their preferred imputation strategy. Certain study designs would certainly favour low coverage sequence versus arrays. From a cost and genotyping accuracy perspective, low coverage sequence should be favoured over genotyping microarrays for most human studies of common disease, as has been shown in our previously QUILT

work, as well as by the GLIMPSE authors. This is driven by the increased accuracy of lc-WGS and imputation versus arrays at current price points, and that the genetic architecture suggests that most causal variants are comparatively common. There are some exceptions in which arrays would be favoured over lc-WGS. In particular, arrays directly type genetic variants. As such, in cases where there is a substantial contribution to the genetic architecture from variants which are hard to impute, in particular rare population specific variants which are poorly captured in large haplotype reference panels, or more generally, variants with little LD to other variants.

The results suggest that QUILT2-nipt could significantly enhance GWAS and PRS performance using imputed fetal genotypes. Could the authors elaborate on the implications for clinical applications, particularly the accuracy requirements in prenatal risk assessments? Are there recommended r^2 thresholds for polygenic score calculations based on fetal fraction? Given the ongoing debate around the clinical utility of PGS and the uncertainty of fetal genotypes (depending on seq depth + FF, see also below), this topic could result in diverse opinions.

Thank you for the comment. For GWAS, we think it is a simpler case, where we envision the data already having been collected. The genotypes would then be imputed with accuracy as expected based on sequencing depth, and the results could be used like a normal GWAS.

For clinical applications using PGS, it is more complicated. As shown in the paper, increasing sequencing depth improves fetal genotype imputation accuracy, and as such, the r^2 for polygenic score calculation - i.e. the squared correlation between the polygenic score from fetal NIPT imputed genome vs a high quality WGS genome. Therefore it should be a matter of cost, not technical possibility, at least for PRS and common variants. As such, the question would then be one of value - whether the costs and benefits of the higher coverage required to ensure a certain reliability of prenatal PRS for the PRS to be useful, outweigh the cost and benefit of not having that PRS but still being able to perform aneuploidy detection. We note that methods which take the uncertainty of genotype imputation into PRS interpretation are emerging, e.g. <https://pubmed.ncbi.nlm.nih.gov/37490908/>.

For clinical diagnoses relying on rare variants, lcWGS based imputation is unlikely to be sufficiently accurate, and in such cases, it is more likely that deep fetal exome sequencing would be preferable.

We have changed the text in the Discussion

From a clinical perspective, relevant traits that are important for health care, such as hyperglycemia, gestational diabetes and birth weight, are genetically influenced by both maternal and fetal genomes 14,43,46,48–50. Therefore, the imputation of the fetal genome before childbirth affords the opportunity to calculate PRS prenatally, which may impact future health risk management if the PRS is sufficiently predictive, though future studies will be required to fully evaluate this potential. Recent papers, which take the uncertainty of genotype imputation into PRS interpretation into account, are of particular relevance here⁵¹. Nonetheless, the imputation of specific variants, in particular rare variants, remains challenging. Therefore it is unlikely to be used for clinical diagnosis, and in that context, recent approaches using non-invasive deep fetal exome sequencing are more promising 22,23. We note that further extension to QUILT2-nipt should still operate on fetal exome sequencing, so in addition to accurate rare variants, such data would also allow for genotype imputation and PRS calculation.

Minor:

The abstract describes the fetal genome imputation results using QUILT2 at 0.25× sequencing coverage with a 10% fetal fraction as “accurate,” yet the r^2 is only 0.465. This statement seems potentially misleading—could the authors clarify this assessment?

Thank you for this comment. We had meant “accurate” to refer to the mother, but this phrasing is unclear. We’ve changed the language to the following.

“... we see accurate imputation of the mother ($r^2 = 0.966$) and modest imputation of the fetus ($r^2 = 0.465$)”

Line 154: Please confirm the minor allele frequency (MAF) levels used to define common SNPs.

We stratify SNPs into bins based on gnomAD allele frequency, and while we show all bins in figures, we focus on the following exemplar bins, as stated in the text. We note that this would give very similar results, and effectively identical qualitative results, had we defined common SNPs otherwise e.g. 5-50%.

As exemplars, we used ‘very rare’ to refer to SNPs with MAF of 0.01-0.02%, ‘rare’ to refer to SNPs with MAF of 0.1-0.2%, and ‘common’ to refer to SNPs with MAF of 10-20%.

There are several typos in the manuscript that need to be fixed.

We apologize for any typos. We have done our best to correct any typos we have found, but please let us know if you find more.

Reviewer #1 (Remarks on code availability):

See above. I think the quality of the Github repo is very very high.

Thank you!

Reviewer #2 (Remarks to the Author):

Summary of the key results, Originality & significance

Li et al. reported an improved method for genotype imputation in data produce from short- and long-read sequencing data, named QUILT2. The methodology is primarily based on their previous work (QUILT) that makes use of sequencing reference data to impute undetermined genotypes of shallow-sequencing data (PMID: 34083788). The authors stated two technical and a methodological improvement over their previously published works (PMIDs: 27376236, 34083788). Technical improvements include: (i) a memory efficient PBWT with the purpose of finding haplotypes in the reference panel that shares long matches with constant computational complexity and “very low” memory footprint; (ii) a two-stage imputation approach that first samples read labels and makes use of common SNPs to find an optimal subset of haplotype reference panel. Methodological improvement includes assumes that three haplotypes are present in the NIPT data.

This reviewer did not find these improvements conceptually novel relevant to their previous works (PMIDs: 27376236, 34083788) and the existing work GMPSE2 (PMID: 37386250). In additions, although the authors claim applicability of their method (three-haplotype option) for non-invasive prenatal testing (NIPT) by discerning fetal and maternal genomes and imputing the fetal genome, in my view, the assumptions are far from real cfDNA component in blood stream of pregnant women (see below). And that it has been performed on simulated data from whole-genome sequencing data of trios. For the latter, the r^2 values are specifically too low to claim that the methodology works on low-pass sequenced NIPT data.

Thank you for taking the time to review our manuscript and for the constructive comments.

Major comments:

- The QUILT2-nipt method assumes the observed sequencing reads are distributed among three haplotypes (maternal transmitted, maternal untransmitted, and paternal transmitted). However, the complexity of real-world data, including recombination in maternal haplotypes and errors in fetal fraction estimation, may introduce inaccuracies that are not addressed. How do you address recombination sites between the maternal homologs?

We'll answer this jointly with the comment below. Also, we analyzed one real NIPT sample during this revision.

- Given my comment above, it is incorrect to assume that NIP data has "triploid nature", it is rather tetraploid (prenatal chromosomes recombine).

Yes this is an issue theoretically and it is one that might negatively impact imputation, but probably less than you think. To expand, you are correct that in the blood of the pregnant mother, if we consider the entire chromosome of origin of any fragment of DNA, than these chromosomes will have four haplotypes: the first haplotype of the mother, the second haplotype of the mother, a mosaic maternal haplotype (derived from crossovers of the maternal haplotypes) and a paternal haplotype. However, if we consider a single arbitrary short stretch on the chromosome, the mosaic maternal haplotype will exactly match one of the two maternal haplotypes, and as such there are only three distinct haplotypes, present at three frequencies. The problem therefore is the scale at which these two truths diverge, and what it means for imputation.

Human recombination is about 1 cM / Mbp, with one obligate crossover during meiosis. As such we would expect on the order of a few recombinations (crossovers) per chromosome, with a higher rate in

the telomeres. With QUILT2, we typically impute in windows that are between about 1 to 5 Mbp. So as an example, on a 100 Mbp chromosome, if there were two maternal crossovers, and if we imputed in 2 Mbp windows with 500 Kbp buffers, we would expect this problem in the central region in about 2 out of 50 windows - in the other 48 windows, there is no error in this key assumption in QUILT2-nipt. In the 2 problematic windows, in the worst case scenario, that the crossover happens in the middle, then we'll incorrectly impute one of the halves. As such, for the imputed maternal chromosome, we would expect imputation problems in about $2 \times 1 \text{ Mbp} / 100 \text{ Mbp}$ or about 2% of the chromosome. We note that this problem would mostly affect the imputed fetal genome (through difficulties in determining the transmitted maternal haplotypes), and would only very minorly affect the imputation of the maternal genome (with both maternal haplotypes).

This assumption could be fixed through an alternate modelling strategy that adds in a new hidden variable that explicitly models which of the two maternal haplotypes is the transmitted one, along the region to be imputed. This is a consideration for a future model, however this would be a substantial change to the model, and the potential benefit of this (improving fetal imputation in $\sim 2\text{-}5\%$ of the genome, or less with shorter windows) would need to be balanced against other potential changes.

In terms of errors in fetal fraction estimation, we showed in the manuscript in Figure 5a that the model is robust to errors in this parameter by giving QUILT2 biased fetal fraction. These errors are in line with, or above, what would be expected from commonly used methods to estimate fetal fraction.

- R2 is very low for real NIPT data. Therefore application of QUILT2 can lead to wrong SNP calls. Could you clarify which parts of genomes cause this low r2 value?

To clarify, we impute/genotype known SNPs in a reference panel, and for many people, the preferred output is a dosage, which is the estimated number of alternate alleles. The r2 is very accurate for the mother, and it's also accurate for the fetus with effective sequencing coverage $> 0.4x$, e.g. $r2 > 0.866$ for $4x$ cfDNA with 10% Fetal Fraction, for common variants (Also, see our response to Reviewer 4 Minor Comment 14 about imputing common variants that causes Mendelian diseases).

Regarding the imputation error, we expect mostly a generally uniform imputation error rate across the genome, however some subtle differences are possible. In particular we expect that non-uniform imputation error across the genome is principally caused by variation in recombination rate, variation in read mapping quality, and variation in haplotype reference panel accuracy (including phasing accuracy), itself also likely related to the previous two points.

For the recombination rate, a higher recombination rate will affect imputation both for the reason given in the previous answer, and as a higher recombination rate will shorten the length of haplotypes copied identically between the reference panel and the focal haplotype to be imputed, making its detection using lc-WGS more difficult. As such in particular the telomeres should have lower accuracy, as should regions of the genome with high or unusual recombination rate and coalescent histories, like the MHC.

For read mapping quality, clearly erroneous read mapping will negatively impact imputation. However, with appropriate minimum mapping quality thresholds, this should be a minor concern. The main concern would likely be from reads from regions of the genome with recent duplications, where one of those duplications is not captured appropriately in the reference genome.

Finally variation in reference panel accuracy, both singleton genotyping errors, and phase switch errors, will affect imputation accuracy, and will be non-uniformly distributed. We expect this to largely follow the same regions as the previous two issues.

- The method has higher accuracy for common variants but its applications with (very) rare variants is challenging and substandard. While this is expected, the reliance on filtering out rare variants early in the iterative imputation process could overlook clinically relevant but less frequent variations. Specially in the context of NIPD. Please clarify how you would deal with this challenge?

Thank you for the comments. Two things here. First, with respect to the iterative process, while the rare variants are not used in the early rounds of imputation, they are used and output at the end. So an output imputation file will contain imputed genotypes at all SNPs in the haplotype reference panel, regardless of frequency.

Second, with regards to clinical applications of rare variants imputed from lc-WGS. Our results clearly suggest that we can not impute rare variants to sufficient accuracy for clinical purposes. If that is the aim, then alternative approaches, such as the deep prenatal whole exome sequencing technology recently developed by Brand, H. et al. (High-Resolution and Noninvasive Fetal Exome Screening. *N Engl J Med* 2023), should be considered. We note that further extension to QUILT2-nipt can work on such data (cfDNA NIPT whole exome sequencing), so in such a case, one could get rare variants directly from the WES, and common variants / PRS across the whole genome using QUILT. We've added to the discussion with these points as follows.

Nonetheless, the imputation of specific variants, in particular rare variants, remains challenging. Therefore it is unlikely to be used for clinical diagnosis, and in that context, recent approaches using non-invasive deep fetal exome sequencing are more promising 22,23. We note that further extension to QUILT2-nipt should still operate on fetal exome sequencing, so in addition to accurate rare variants, such data would also allow for genotype imputation and PRS calculation.

- The authors discuss the potential clinical utility of imputed fetal genomes for traits such as gestational hypertension and preeclampsia but does not provide sufficient empirical evidence or validation studies to support these claims. Without these and my concerns above, this reviewer finds such phrases more over-statements.

Thank you for the comments. We agree that this is an over-statement for gestational hypertension and preeclampsia, where the contribution of the fetal genome is not supported by the paper (PMID: 36947680) we're referencing. However, there are many papers showing how both the mother and fetal genomes contribute to the birth weights and fetus growth, which are also linked to maternal hyperglycemia and gestational diabetes (ref: PMID: 29463506, 31043758, 32841251, 31043758, 37798380, 38297123). Specifically, the findings by Chen et al. 2020 (<https://pubmed.ncbi.nlm.nih.gov/32841251/>) concluded that, "Alleles elevating blood pressure are associated with shorter gestational duration through a maternal effect and are associated with reduced fetal growth through a fetal genetic effect. Alleles that increase blood glucose in the mother are associated with increased birth weight, whereas risk alleles for type 2 diabetes in the fetus are associated with reduced birth weight."

Therefore, we discuss potentials for these traits instead and note that future studies will be required to fully evaluate these potentials with the following changes in text.

*In the second paragraph of Discussion, the original sentence was
From a clinical perspective, as we move toward leveraging PRS in health risk management, the imputation of the fetal genome before childbirth should afford the opportunity to change clinical practice for relevant traits, such as gestational hypertension and preeclampsia.*

The new sentence is

From a clinical perspective, relevant traits that are important for health care, such as hyperglycemia, gestational diabetes and birth weight, are genetically influenced by both maternal and fetal genomes 14,43,46,48–50. Therefore, the imputation of the fetal genome before childbirth affords the opportunity to calculate PRS prenatally, which may impact future health risk management if the PRS is sufficiently predictive, though future studies will be required to fully evaluate this potential.

- The authors briefly state the potential of QUILT2 to identify structural variations and copy number variants but did not provide how or a benchmarking for these applications. Please clarify.

As is standard, we have included in the Discussion areas of potential future methodological improvement. This capability does not yet exist in QUILT2, and it is therefore not benchmarked. We chose to highlight this as the methodological changes would be difficult but reasonable, and the potential clinical impact high. Specifically, there are already clinical tests that offer sub-chromosomal CNV calls, e.g. Roche Harmony for 22q11.2 microdeletions, however with limited sensitivity at reasonable specificity. We think it is worth investigating whether an approach based on QUILT2 (which partitions reads) would be more accurate.

For further discussion on this, including discussion of our ongoing work in this area, please see our response to Reviewer 4, Minor Comment 15.

- For the diploid imputation performance, the authors indicate that they “used Chr 20 for convenience”. Why did you make such choice? Is this then correct that all the data presented in this manuscript are using Chr 20 only?

It is generally standard in imputation benchmarking to use a single chromosome for extensive testing. This makes analysis faster when there are multiple methods and parameters to evaluate. It is also reasonable as while imputation accuracy varies along the genome (see above) it generally does not vary appreciably between chromosomes. Similar studies like the GLIMPSE/GLIMPSE2 paper do similar things when comparing methods and parameter choices.

However, we do evaluate the performance for the whole genome in some analysis, e.g. we imputed the whole genome to conduct the GWAS/PRS. In addition, in this revision, we analyzed a real NIPT sample and evaluated the performance of QUILT2-nipt vs QUILT2-diploid by imputing the whole genome as well.

- The authors devised a 32-bit grid strategy and allowed transition after 32nd pair of SNPs. It is not clear to this reviewer why 32 and how the transitions have happened?

This technical detail is the same as for the previously published QUILT (QUILT1) manuscript. That being said, the 32 number is chosen for computational reasons. We restrict ourselves to bi-allelic SNPs so haplotypes can only have reference or alternate bases i.e. 0's (reference) or 1's (alternate), and as such, 32 consecutive SNPs in one reference haplotype can be encoded with a single 32 bit integer, which is highly efficient. Transitions (recombinations) are then forced to happen only every 32 SNPs. This is a deficiency, as recombinations might occur between those 32 SNPs, however, in the context of WGS reference panels, where SNPs are overwhelmingly rare, and recombinations infrequent, occurring many thousands or tens of thousands of SNPs apart, it has at most very modest effects on accuracy, while substantially speeding up imputation.

- Description of QUILT2 for (ONT) long-read sequencing sounds preliminary. Please clarify how you make use of long-read sequencing data and kb-Mb size haplotypes. How it differs from short-read

sequencing? How it performs for structural rearrangements? How do you potentially detect erroneous SNVs of the ONT data?

To reiterate first, we do not discover novel variants including structural variants. The technical details of how QUILT2 can handle long reads are broadly the same as in the QUILT1 paper, and as such, have been covered before. Here we mostly use the long read analysis to reaffirm this capability with QUILT2.

In addition, as is standard, we discussed how it can be useful for future method developments. For example, most SV methods are alignment based, and suffer as the true underlying phase of the reads is unknown - it would be easier to genotype them if we could phase them into their underlying haplotypes. In this field, people normally use the "whatshap haplotag" program to accomplish it, however this requires high coverage sequencing data. This could be done using QUILT2, with lower coverage data. More generally we could envision an iterative process of underlying haplotype proposal, alignment, and read phasing, using QUILT2 and other methods.

For further discussion on this, including discussion of our ongoing work in this area, please see our response to Reviewer 4, Minor Comment 15

Minor comments:

- In the beginning of the Results section "...QUILT2 is based upon, and improves, our previous model QUILT1 for diploid genotype imputation, which operates, which uses an iterative approach...". Either use "which operates" or "which uses"

Thank you. We've corrected the language, i.e. "which uses" was removed.

- Please make figures that illustrate QUILT2, clearer. Specially for the grid search and using common SNPs as anchors.

We appreciate your comment, however there are a lot of key ideas to get across in QUILT2, namely the two technical improvements (msPBWT & using common SNPs as anchors), as well as the methodological improvement (NIPT). It is difficult to get all three across in a single figure without being overwhelming. We chose to focus Figure 1 on the msPBWT and NIPT ideas, and the link between NIPT and diploid. We have two Extended online figures that cover msPBWT (Extended Online Figure 1) and the anchor SNP idea (Extended Online Figure 2) in high detail (these took a lot of time to make!). We hope they are sufficient for the average reader. We note that the caption for Figure 1 includes the following, to direct the reader to these figures

More details about the msPBWT algorithm are shown in Extended Online Figure 1. Details about the iterative approach using common and all SNPs, which is not shown in this schematic, is shown in Extended Online Figure 2.

- The authors state that "In some countries, NIPT is routinely offered to all pregnant women, for instance in the Netherlands, with an average uptake rate of 46%". This is not up to date. This uptake rate is from the TRIDENT study that was finalized in March 2023. Since April NIPT in the Netherland is a screening program with very high uptake rate. Please correct.

Thank you for the comments. We could not find a more up to date reference for this figure, in either English or in Dutch. We have additionally added a reference to a number for Belgium. The new sentence is

In some countries, NIPT is routinely offered to all pregnant women, with recent uptake rates at around half the population or more, with an uptake rate of 79% and 46% for Belgium (79%) and Netherlands (46%) respectively 17.

We would happily update this further if provided alternative references to use.

- Please insert page and line numbers to make the review process easier and more efficient.

Thank you for this nice suggestion. We've now added the page and line numbers.

Reviewer #2 (Remarks on code availability):

There are different GitHub repositories referred in "Code availability" section of the manuscript. Please make a clear scheme how they are related to each other.

We deleted some redundant repositories, and reiterated the functionality of each repository. QUILT2 is the main focus of the manuscript and is listed first. The second repository is the code used to run the benchmarking in the manuscript.

The codes require extensive documentations, indicating what function or piece of code does what. For instance, "## known ped, weird format" is not really informative.

QUILT2 has a README with instruction and usage information and examples of how to use it. QUILT2 has a defined user-interface after installation which is heavily documented. We do not expect the users to use internal QUILT2 functions so we have not put the same care into documenting them.

Reviewer #3 (Remarks to the Author):

The article presents QUILT2, a method for genotype imputation from low-coverage whole-genome sequencing (lc-WGS), building on the QUILT1 framework. QUILT2 integrates a memory-efficient implementation of the positional Burrows-Wheeler transform (msPBWT) and a two-stage imputation process, aiming to improve the speed and scalability of imputation. Additionally, it introduces a novel approach for imputing both maternal and fetal genomes from non-invasive prenatal testing (NIPT) data. While the method shows promise in some applications, especially for NIPT data, the manuscript raises several concerns.

Thank you for taking the time to review our manuscript and for the many constructive comments.

Biased benchmark

Overall, the benchmark puts emphasis on QUILT2 outperforming GLIMPSE2, while it seems to avoid showing scenarios where GLIMPSE2 may be preferred. For example, it is unclear if GLIMPSE2 demonstrates better performance at higher coverage levels, as the benchmarking ends at 2x coverage, where differences seem to favor GLIMPSE2. This omission limits a comprehensive understanding of QUILT2's strengths and weaknesses. Additionally, computational time is reported at 0.25x coverage, where GLIMPSE2 is shown to be about four times faster, but it is unclear if this

difference is maintained at higher coverage. A more comprehensive and fair evaluation of methods across coverage levels and contexts would enhance the manuscript's scientific validity.

Thank you for the comments. We appreciate this and have reworked the benchmarks and their presentation to try and be more fair to both methods. We have added 4.0x coverage into the benchmarking.

In addition, in the text, we have acknowledged the better performance of GLIMPSE2 including for rare variants at higher coverage. The original sentence was:

QUILT2 was more accurate than GLIMPSE2 at sequencing coverage $\leq 0.5\times$, regardless of the reference panel, and particularly at $0.1\times-0.25\times$.

The new sentence is:

QUILT2 was more accurate than GLIMPSE2 at sequencing coverage $\leq 0.5\times$ regardless of the reference panel, while GLIMPSE2 was more accurate than QUILT2 for rare variants at coverage $\geq 2.0\times$

We have also tried to better emphasize differences in reference panels. The original sentence was:

No method was uniformly superior for these rare SNPs at higher coverage samples, with QUILT2 being slightly more accurate than GLIMPSE2 for the larger panel (UKB) but not for the smaller panels (KGP and HRC)

The new sentence is:

For imputation of modern samples with smaller panels (KGP and HRC), we found that QUILT2 is less robust and accurate than QUILT1 and GLIMPSE2 for rare variants with default settings (Supplementary Figure 1)

In addition, regarding the computational time, we apologize that we forgot to write the coverage of samples used in Figure 3A, which is 1.0x instead of 0.25x as the reviewer mentioned. We now updated Figure 3 to make this more clear, as well as including the cost on RAP for coverage 2.0x as the reviewer requested in another comment. About the broader point, about computational run time as a function of sequencing depth, this is now covered through Figure 3B, where we show cost, driven by runtime, as a function of sequencing depth. QUILT2 is faster for lower coverage samples than higher coverage samples, as some expensive calculations in the HMM can be skipped if windows of 32 SNPs have no reads in them, and more generally, as there are fewer reads to iterate over in the HMM.

Lack of citations to previous work

The authors seem to overstate the novelty of their approach by not clearly citing relevant prior work underlying QUILT2. Several key computational features of the QUILT2 method have been previously introduced by other methods, yet there is a noticeable lack of acknowledgment of these contributions.

The msPBWT shares a lot of similarities with the multi-allelic PBWT (Naseri et al, 2019) and/or Syllable-PBWT (Wang et al, 2023).

PBWT for haplotype selection has been used in numerous methods already: EAGLE2, SHAPEIT4, SHAPEIT5, IMPUTE5, GLIMPSE1, and GLIMPSE2.

Treating rare and common variants differently to speed up computation has been used in BEAGLE5, IMPUTE5, SHAPEIT5, and GLIMPSE2.

Also, not discussing methodological differences between GLIMPSE2 and QUILT2 is a significant oversight given the similarities between the two methods. The first mention of GLIMPSE2 occurs in the method comparison and does not provide any context for this comparison.

Thank you for the comment. You are right that we have not done a sufficient job referencing the literature and previous methods, for which we apologize.

We note that Syllable-PBWT, b-PBWT (original “binary” PBWT from Richard Durbin) and IMPUTE4 were referenced in the section of “Haplotype matching with msPBWT” in the Supplementary Note, however the reference page was somehow not included when we compiled the Supplementary Note previously, a problem which has now been fixed.

We now acknowledge the relevant works. To begin, we have substantially re-worked the second paragraph in the introduction, to offer a brief introduction to QUILT as well as GLIMPSE, as well as what motivated GLIMPSE2.

Previously, we developed QUILT for rapid genotype imputation of lc-WGS data with a reference panel 4. Around the same time, Rubinacci et al. introduced GLIMPSE, another method for genotype imputation 4,5 . Both QUILT and GLIMPSE share a core two-stage imputation framework, where per-sample imputation leverages a small set of conditioning reference haplotypes. For haplotype matching, QUILT employs the Li and Stephens model 9, which is more robust, whereas GLIMPSE relies on the positional Burrows-Wheeler transform 10 (PBWT), which offers computational efficiency. QUILT and GLIMPSE further differ in how they treat input data, with GLIMPSE using per-SNP genotype likelihoods, while QUILT uses and phases the sequencing reads directly, which is preferable for linked reads or long-read data. Overall, QUILT is at least as accurate as GLIMPSE and more robust, however GLIMPSE tends to be faster 5. Since the publication of these methods, reference panels have expanded significantly in both the number of haplotypes as well as variant density, for example many studies now use the UK Biobank whole-genome sequencing (UKB WGS) 11. To accommodate these growing panel sizes, the authors of GLIMPSE developed GLIMPSE2 6. GLIMPSE2 focuses on improving speed and facilitating biobank sized reference panels by incorporating more sparse data structures for both speed and memory efficiency, as well as the introduction of a dedicated version of PBWT, termed sparse PBWT.

In addition we better reference the history of the key ideas of QUILT2 in the results, first with msPBWT

The use of PBWT is common for haplotype identification in genotype phasing and imputation 5,6,29–31, including with GLIMPSE and GLIMPSE2, and derivatives exist that work on non-binary symbols, such as multi-allelic and Syllable PBWT 32,33.

Second, for the use of rare and common variants

As such, several methods for genotype phasing and imputation differentiate between common and rare SNPs, for both speed and RAM reasons, including GLIMPSE2 and other methods 29,31,34.

Presentation of results

I believe the whole Figures 2 and 3 should be carefully rethought and redesigned for clarity and more importantly for transparency.

We appreciate the constructive comment. We have re-designed Figures 2 and 3.

- Figure 2a: The comparison of methods for coverage > 0.25x is simply not visible.

We have adjusted the y-axis so that the difference is more visible. We have also added in the 4x benchmarking.

- Figure 2c: The choice of 0.25x coverage is unclear. It seems that the difference between methods do not really differ to what is already shown in 2a. What does the comparison look like for 1x coverage?

0.25x was chosen because the data was only collected at slightly above 0.25x coverage. However, since the haplotagged data is not in common use, and to facilitate overall greater readability of this figure, we have removed this analysis from Figure 2.

- Figure 2d: Accuracy is presented differently for this analysis compared to 2a-c. That makes results harder to read.

We made that choice because we followed the design of GLIMPSE2 paper, i.e use non-reference concordance as y-axis and use sequencing coverage for x-axis. We've now made the x-axis the same as the other figures, but we stick to the non reference concordance rate for the y-axis because it's more informative than the r2 of genotype dosages for ancient DNA analyses.

- Figure 3a-b: These two figures need to show computing time for datasets comprising thousands of samples and even more, cases where compute time and memory usage are a real issue.

We'll answer this jointly with the same comment below in "Lack of major common use cases".

- Figure 3c: Why #samples=8 here and why stop at 1x while most plots show results up to 2x?

We now added 2x. We will give a more complete answer to why we compare N = 8 for QUILT2 vs N = 100 for GLIMPSE2 for cost reasons in the section below "Lack of reference for major common use cases".

Finally, it seems unfair to choose the reference panels for which QUILT2 performed the best in some panels of fig. 2: 2a shows results for UKB, while 2d for 1000G. The authors should either show results for all reference panels as the main figure or pick one and stick to it.

Thank you for the comment. We now focus primarily on the UKB panel.

Lack of reference for major common use cases

Major low-coverage WGS initiatives, like Blended Genome-Exome (BGE), focus on higher sequencing depths (2-4x), yet the manuscript does not address QUILT2's application at these depths for large-scale cohorts. I think all benchmarks should at least include 4x coverage configuration beside the coverages already presented.

Thank you for the comment. We've now added 4x in the benchmarking and re-designed the Figure 2.

The timing benchmark seems also to miss the point: compute time for single-sample imputation is not anymore an issue. Scalability for hundreds of thousands of samples as in typical GWAS is actually the critical aspect. This is not addressed in the benchmark, nor in the discussion. Readers will want to

know how running times and RAM usage scale on large datasets comprising many thousands of samples.

We appreciate this suggestion, and we partly agree. However a key consideration of current analyses is that large reference panels increasingly live on the cloud (e.g. the UKB WGS), and can't be run locally. The cheapest way to run jobs on the cloud is to use low priority, preemptible jobs. On the UKB RAP, high priority jobs are about three times more expensive. As such it makes sense to run jobs with fewer samples so they run quickly. In the GLIMPSE2 paper, in their supplement, we note *In practice, we found that jobs that run on large machines (such as "mem3_ssd1_v2_x32") are interrupted and executed "on-demand" after ~35 minutes of computation.*

We saw similar results recently as shown in the below screenshot

Name	State	Executable	Launched By	Launched On	Started Running	Duration	Cost	Priority
cost_glimpse2_1.0x_n100_010	Done	Swiss Army Knife (v4.1...	Zilong Li	Mar 29 2025, 10:32 AM	Mar 29 2025, 11:05 AM	37m	£0.0887 Final	high
cost_glimpse2_1.0x_n100_009	Done	Swiss Army Knife (v4.1...	Zilong Li	Mar 29 2025, 10:32 AM	Mar 29 2025, 10:32 AM	39m	£0.0217 Final	normal
cost_glimpse2_1.0x_n100_008	Done	Swiss Army Knife (v4.1...	Zilong Li	Mar 29 2025, 10:31 AM	Mar 29 2025, 10:32 AM	36m	£0.0202 Final	normal
cost_glimpse2_1.0x_n100_007	Done	Swiss Army Knife (v4.1...	Zilong Li	Mar 29 2025, 10:31 AM	Mar 29 2025, 10:32 AM	35m	£0.0196 Final	normal
cost_glimpse2_1.0x_n100_006	Done	Swiss Army Knife (v4.1...	Zilong Li	Mar 29 2025, 10:31 AM	Mar 29 2025, 10:32 AM	36m	£0.0200 Final	normal
cost_glimpse2_1.0x_n100_005	Done	Swiss Army Knife (v4.1...	Zilong Li	Mar 29 2025, 10:31 AM	Mar 29 2025, 10:32 AM	55m	£0.0307 Final	normal
cost_glimpse2_1.0x_n100_004	Done	Swiss Army Knife (v4.1...	Zilong Li	Mar 29 2025, 10:31 AM	Mar 29 2025, 10:31 AM	56m	£0.0312 Final	normal
cost_glimpse2_1.0x_n100_003	Done	Swiss Army Knife (v4.1...	Zilong Li	Mar 29 2025, 10:31 AM	Mar 29 2025, 10:31 AM	39m	£0.0219 Final	normal
cost_glimpse2_1.0x_n100_002	Done	Swiss Army Knife (v4.1...	Zilong Li	Mar 29 2025, 10:31 AM	Mar 29 2025, 11:03 AM	39m	£0.0919 Final	high
cost_glimpse2_1.0x_n100_001	Done	Swiss Army Knife (v4.1...	Zilong Li	Mar 29 2025, 10:31 AM	Mar 29 2025, 10:31 AM	36m	£0.0202 Final	normal
cost_glimpse2_1.0x_n100_000	Done	Swiss Army Knife (v4.1...	Zilong Li	Mar 29 2025, 10:31 AM	Mar 29 2025, 11:03 AM	37m	£0.0872 Final	high

This screenshot shows that 3 out of 11 jobs of GLIMPSE2 were interrupted when run with N=100. For those preempted jobs, we had to rerun several times to be able to finish them normally, with a corresponding increase in cost. Note that all costs reported in the paper are idealized, i.e. cost of the least expensive machine times the amount of time used, as done in the GLIMPSE2 paper.

Both QUILT2 and GLIMPSE2 do not use information from other samples when imputing, and as such, can be completely parallelized per-sample. In the supplement to the GLIMPSE2 paper, they show they achieve optimal per-sample efficiency with $N \approx 100$ samples, so it would make sense to target that sample size in any given job. QUILT2 by contrast has no such per-sample efficiency costs so would presumably be run in very small jobs e.g. 8 samples like we showed before (a multiple of the number of available 4 cores) as we've previously highlighted, as downstream tasks like merging VCFs are very inexpensive when done correctly. As such, the issue of running large datasets with many thousands of samples is really just an issue of how to most efficiently run per-sample, which is why we've focused as we have on comparatively small sample sizes like shown in Figure 3.

Ideally, if the cloud platform is not overloaded and all jobs with 1h runtime can be expected to finish in normal priority, it makes sense to run QUILT2 with $N = 32$ samples. Here are the costs for different numbers of samples as an input batch (as always, the idealized cost, assuming there is no pre-emption).

	QUILT2(N=8)	GLIMPSE2(N=8)	QUILT2(N=32)	GLIMPSE2(N=100)
0.1x	0.2603240	0.3202330	0.2373724	0.09143739
0.5x	0.3284220	0.3111820	0.2994665	0.08885302

1.0x	0.4072600	0.3138550	0.3713537	0.08961626
2.0x	0.5210532	0.3321573	0.4751142	0.09484219

As such, we've now amended the Figure and the text in the benchmark to reflect these numbers (N = 32 for QUILT2, N = 100 for GLIMPSE2), as these are reasonable sample sizes that someone might choose for arbitrarily large GWAS.

Since both QUILT2 and GLIMPSE2 do not borrow information across samples, and as the cheapest way to run jobs on the RAP is using low or normal priority jobs which can be interrupted 6, it is advantageous to impute large datasets using small batches, and then to combine results afterwards. In their previous work, the authors of GLIMPSE2 showed that optimal per-sample runtimes were achieved at approximately 100 samples 6. Comparing per-sample costs for QUILT2 (using N = 32) and GLIMPSE2 (using N = 100) we see both methods are inexpensive though QUILT2 (£0.237, £0.299, £0.371 and £0.475 for 0.1x, 0.5x, 1.0x and 2.0x respectively) is more expensive than GLIMPSE2 (approximately £0.09 for all coverages). We note these are idealized estimates, and real costs will depend on prices at run time, server congestion, etc, and furthermore, that default parameter settings for QUILT2 favour accuracy over speed.

We further make the following mention in the Discussion

In terms of speed, QUILT2 is slower and less efficient than GLIMPSE2 when run at scale, however it is faster when using very large reference panels to process a small number of samples.

Ancient DNA benchmark

The authors do not describe how imputation has been run on ancient DNA. Established pipelines involving GLIMPSE are already used in real-world ancient DNA applications, with GLIMPSE2 authors explicitly advising against its simplistic genotype likelihood model for complex scenarios like aDNA. If the authors choose to retain the ancient DNA benchmark, they should compare their tool with the recommended pipeline outlined by Mota et al., Nat Comm. 2023.

Thank you for the comment. In the initial submission we did use the recommended pipeline from Mota et al Nat Comm 2023. We've made this more clear by mentioning this in the Results section where we say

We also assessed the imputation performance on aDNA from Afanasievo culture (~4.6kya) with three reference panels. We followed the Mapache 38 pipeline used in the GLIMPSE2 paper, which implements the practice by Mota et al. 2023 7.

We use the Mota et al pipeline to generate the BAM files used by QUILT2 and GLIMPSE2 directly. We note in the Mota et al 2023 paper that they show that corrected methods for genotype likelihood calculation are only marginally more accurate than using a simplistic model anyway.

We also lightly re-wrote the methods. The original text said

Then we used the mapache ²³ snakemake workflow, to map the FASTQ files against the human reference genome GRCh38 to generate the BAM files for imputation benchmarking

While the new sentence reads

Then we used the mapache ²³ snakemake workflow, which implements the best practice for aDNA imputation ⁷, to map the FASTQ files against the human reference genome GRCh38 to generate the BAM files for imputation benchmarking.

Long reads

The application of QUILT to long reads is a key advantage, but this was demonstrated in prior work. It is unclear why similar results are highlighted in a main figure here or what this benchmark aims to achieve. If the focus is on long-read performance, the comparison should emphasize improvements over QUILT1, rather than contrasting with GLIMPSE2, which is not designed for long reads.

Thank you for the comment. Indeed there are two factors at play: QUILT1 (small panel) vs QUILT2 (large panel); and QUILT2 (long read favourable method) to GLIMPSE2 (less favourable to long reads). It feels worthwhile to evaluate both changes simultaneously, which we now do, in Figure 2C. While this generally highlights the performance of QUILT2 using UKB, QUILT1-HRC is more accurate than QUILT2-UKB at 0.5x coverage. Nonetheless as far more SNPs are imputed using the UKB panel, this would undoubtedly still prove a draw for most users.

The re-worked Results paragraph is as follows

Lastly, we evaluated the imputation accuracy using long sequencing reads from Oxford Nanopore Technologies (ONT) by calculating both the r^2 at variant level and F1-score at sample level. To investigate the gain of using the UKB panel for long-read imputation, we included QUILT1 with HRC for comparison. We found that QUILT1-HRC performed better than QUILT2-UKB for coverage $<1.0x$ but with fewer variants imputed (Figure 2c). Additionally, we evaluated QUILT2 and GLIMPSE2, the inexpensive and scalable methods that can use the UKB panel. Across all coverages and reference panels, QUILT2 was substantially more accurate compared to GLIMPSE2 for long reads (Figure 2c, Supplementary Table 3), e.g common $r^2=0.937$ vs $r^2=0.695$ at $1\times$ coverage.

NIPT analysis

We find this part of the manuscript useful to the community, nicely carried over and interesting. This is a significant advance in the field.

Thank you for this positive comment. We note that as part of this review process, we have also added in the analysis of a real NIPT sample in Extended Figure 4.

Minor comments

- The imputation performance of QUILT2 at indel sites is not clear. Could the accuracy for SNPs and indels be separated?

Thank you for the comments. QUILT2/QUILT1 don't support indel imputation. We have not benchmarked the performance for indels.

- The statement "For both human and animal studies, QUILT has been shown to be the most accurate and robust approach when a reference panel is available" is overly assertive. Both the GLIMPSE2 study and this manuscript suggest minimal performance differences between methods.

Thank you for the comment. We have toned down this statement, replacing it with

Overall, QUILT is at least as accurate as GLIMPSE and more robust, however GLIMPSE tends to be faster.

- The claim "QUILT2 should retain greater accuracy over GLIMPSE2 in various non-human settings, as the effective population size, and hence SNP density, is often much higher than in humans" requires further clarification. Where is the data showing this?

We have changed this to "QUILT2 should be more accurate than GLIMPSE2 in instances where SNP heterozygosity is high, as would be expected in species with large effective population sizes". If the reviewer would like, we could further amend or remove it.

We do not have data specifically to verify this, however believe it is a reasonable extrapolation. We showed in the paper that QUILT2 retains increased accuracy over GLIMPSE2 for long read sequencing in humans. The difference between using long and short reads is that with long reads, the expected number of SNPs per read is increased. This is akin to using short reads with much higher SNP density. For instance in *Anopheles*, heterozygosity is ~1.5%, vs 0.1% in humans (from the paper Genetic diversity of the African malaria 1 vector *Anopheles gambiae*).

- The statement "We also showed the extent to which large haplotype reference panels can improve both phasing and imputation performance" does not represent a major contribution of this work. This has been demonstrated previously when the reference panel was introduced (citation missing), and this study largely reaffirms those findings.

We had meant this in the context of the paper. While others have shown this, it was important to show in our work, as, for instance, the accuracy of the NIPT imputation, which is more difficult due to the lower coverage and three haplotypes, depends on it. Nonetheless, we have removed this sentence.

- References for some reference panels (HRC, UKB, etc ...) are missing.

Thank you for the comments. We've added these references in the main text at their first mention in the results section on diploid genotype imputation.

Reviewer #3 (Remarks on code availability):

The code is both accessible and functional. Additionally, the documentation is comprehensive and thorough.

Thank you.

Reviewer #4 (Remarks to the Author):

The authors introduce QUILT2, a method that builds on the original QUILT1 for accurate genotype imputation using reference panels. QUILT1 has a linear computational complexity that scales with reference panel sizes and number of SNPs, both of which are becoming increasingly large, thereby slowing the method runtime. QUILT2 addresses this with a scalable, memory-efficient design for low-coverage WGS imputation. Notably, it also enables imputation of fetal versus maternal genomes from cell-free DNA in non-invasive prenatal tests—an underexplored application. The paper effectively demonstrates the method's robustness and efficiency, though a few comments are provided below.

Thank you for taking the time to review our manuscript and for the positive comments.

Major comments:

1. Regarding framing, NIPT is a useful application in its own right, but the framing that it will substantially improve the ability of GWAS to reach millions of participants is somewhat of a funny one, particularly as most phenotypes studied by GWAS are adult-onset diseases or adult quantitative traits. Maybe reconsider NIPT framing as contributing substantially to GWAS in the introduction? Relatedly in the discussion, it is not clear how having common variants from a fetus would change clinical practice in cases of maternal preeclampsia or gestational hypertension, so perhaps clarifying or removing would be helpful.

Thank you for the comments. About the potential for NIPT to contribute substantially to GWAS, we are imagining the GWAS being done on the mother, using their imputed genome, with QUILT-nipt leading to more accurate imputation, and hence more powerful GWAS. We envision the collection of either survey data, or more ideally linking to EHR, to capture adult quantitative traits and binary traits. This could include (through EHR) things like lab measures, as well as simple traits like height and (pre-pregnancy) weight, as well as things like educational attainment, which are important traits in the field of genetics broadly. For binary disease traits, yes there are some traits, for example heart attacks and breast cancer, that are well studied, whose onset is typically after female maternal age, so would be poorly captured directly, though would be amenable to gwas-by-proxy. However there are many traits of interest with much earlier onset including Type 1 Diabetes, neuropsychiatric disorders (e.g. depression and schizophrenia), auto-immune (IBD, RA), epilepsy, etc, that would make GWAS conducted on phenotypes collected from women of maternal age very valuable.

To better rationalize GWAS in the Introduction we have added the following about the value of the phenotypes that can be collected in pregnant women

One option is to utilize DNA and phenotypes from pregnant women. Electronic health records (EHR) usually store rich phenotypes for pregnant women, which could include quantitative traits like height, weight, hyperglycaemia 14, as well as socioeconomic traits like educational attainment, as well as disorders with age of onset before maternal age or during pregnancy, such as neuropsychiatric conditions like schizophrenia, as well as gestational diabetes mellitus 15 and intrahepatic cholestasis of pregnancy 16. A natural way to collect DNA from pregnant women is using cell free DNA from non-invasive prenatal testing (NIPT), which is a sensitive and specific screening test which tests for the common fetal aneuploidies of trisomies 13, 18 and 21, and that, according to the American Society of Obstetricians and Gynecologists (ACOG, 2020), should be offered for all pregnancies regardless of risk.

And further rationalized the potential for GWAS in NIPT by specifically highlighting existing GWAS using NIPT

In 2018, it was suggested that NIPT had already been performed on ten million pregnant Chinese women 18. Further, NIPT has already been used as a successful means of conducting GWAS, with NIPT based GWAS discovering novel associations for intrahepatic cholestasis of pregnancy 16 , gestational diabetes mellitus 15, metabolites 19, glycemc traits 20 and neonatal phenotypes 21.

About the discussion, we have re-written the text surrounding potential clinical applications, and discuss this in response to other comments. We have put the re-written text below.

From a clinical perspective, relevant traits that are important for health care, such as hyperglycemia, gestational diabetes and birth weight, are genetically influenced by both maternal and fetal genomes

39,42,44–47. Therefore, the imputation of the fetal genome before childbirth may afford the opportunity to change clinical practice as we move toward leveraging PRS in health risk management. Recent papers, which take the uncertainty of genotype imputation into PRS interpretation into account, are of particular relevance here⁴⁸. Nonetheless, the imputation of specific variants, in particular rare variants, remains challenging. Therefore it is unlikely to be used for clinical diagnosis, and in that context, recent approaches using non-invasive deep fetal exome sequencing are more promising^{15,16}. We note that QUILT2-nipt should still operate on fetal exome sequencing, so in addition to accurate rare variants, such data would also allow for genotype imputation and PRS calculation.

2. Relatedly, is using fetal genomes in a GWAS / PRS feasible at current accuracy rates? Variants at 1% MAF are only a little over 60% accurate even at 4x coverage and 0.3 FF. Could it introduce false positives? Most of the significant hits are only nominally significant both for mother and fetus (Fig 5b). Similarly, with the current state of PRS as predictors for traits, does matching array data performance with cfDNA really change the feasibility of its use in the clinic? This is an interesting and valuable analysis, but some text in the discussion addressing this caveat could be merited. Could include the study “Novel insights into the genetic architecture of pregnancy glycemic traits from 14,744 Chinese maternities” by Z. Zhu et. al. (2024, Cell Genomics) in the discussion relating to using NIPT data in GWAS.

Thank you for the comment. We’ll answer minor points first before addressing the larger issue. First the specific quoted accuracy, you’re highlighting 0.01%, for variants at 1%, it is closer to 0.80 accuracy. Second, with respect to Figure 5b, this figure shows the relative equivalency of different NIPT coverages versus arrays. For a real PRS, people would use standard PRS methods like clumping and thresholding or LDpred, which may or may not use variants with marginal p-values below genome-wide significance (most do). Third, semi-related, methods are emerging that take the uncertainty of genotypes into account, e.g. <https://pubmed.ncbi.nlm.nih.gov/37490908/>, which would increase the benefit of using imputation genotypes.

For the larger issue, about whether matching array data performance with cfDNA really changes the feasibility of its use in the clinic. Today, the answer is likely no. However, as GWAS sample sizes increase (hopefully facilitated by this study!), and as we get more accurate methodology for building PRS, so too should the accuracy of array data (and cfDNA) based PRS. We think this study, as the first dedicated method for genotype imputation and hence PRS calculation from lc-WGS cfDNA NIPT, plays an important role in showcasing the utility of this data type for making PRS available for eventual potential clinical applications. We have modified the text of the Discussion and specifically highlighted in this response that future studies will be required to properly evaluate the potential of prenatal PRS.

We’ve re-worked the discussion part as follows:

From a clinical perspective, relevant traits that are important for health care, such as hyperglycemia, gestational diabetes and birth weight, are genetically influenced by both maternal and fetal genomes^{14,43,46,48–50}. Therefore, the imputation of the fetal genome before childbirth affords the opportunity to calculate PRS prenatally, which may impact future health risk management if the PRS is sufficiently predictive, though future studies will be required to fully evaluate this potential. Recent papers, which take the uncertainty of genotype imputation into PRS interpretation into account, are of particular relevance here⁵¹. Nonetheless, the imputation of specific variants, in particular rare variants, remains challenging. Therefore it is unlikely to be used for clinical diagnosis, and in that context, recent approaches using non-invasive deep fetal exome sequencing are more promising^{22,23}. We note that further extension to QUILT2-nipt should still operate on fetal exome sequencing,

so in addition to accurate rare variants, such data would also allow for genotype imputation and PRS calculation.

Minor comments:

1. The manuscript would benefit from a careful read through to streamline and improve readability as there are some very complex sentences that are hard to parse. Some examples:

Thank you for the suggestions. We've now corrected many typos and updated the text

a. The first sentence of results lacks clarity: QUILT2 is based upon, and improves, our previous model QUILT1 for diploid genotype imputation, which operates, which uses an iterative approach, as follows. Suggest to remove "which operates" from the sentence.

Thank you. We removed "which operates"

b. There are two i.e.'s in this 55-word sentence. It would be more readable if broken up or streamlined: "However, state-of-the-art imputation methods for lc-WGS are designed for diploid samples, i.e. where the sequencing reads come from two haplotypes 3–6, and there is no dedicated method designed for utilizing the triploid nature of NIPT data, i.e. where the sequencing reads come unequally from the maternal transmitted, maternal untransmitted, and paternal transmitted haplotypes."

Thank you. We have removed "i.e."s and updated the sentence.

However, state-of-the-art imputation methods for lc-WGS are designed for diploid samples, and there is no dedicated method designed for NIPT data taking both the fetus DNA and mother DNA into account.

c. "For diploid samples, we used the WGS derived UKB panel to show that QUILT2 is approximately as accurate as QUILT1, but substantially faster, and uses much less memory, and as such, should be much less expensive in cloud based applications." Example streamlined suggestion: For diploid samples, the WGS-derived UKB panel showed that QUILT2 is as accurate as QUILT1 but significantly faster, more memory-efficient, and cost-effective for cloud-based applications.

We adopted your version. Thank you!

d. "In addition, while QUILT2 is slower than GLIMPSE2 when processing many samples, it is faster, by an order of magnitude on very large reference panels, when processing only a very small number of samples." Suggestion: While QUILT2 is slower than GLIMPSE2 for many samples, it is an order of magnitude faster when using very large reference panels to process a small number of samples.

We adopted your version. Thank you!

e. Line 99-101: suggest that the sentences "Because of the second part of the iterative process, QUILT1 has linear time complexity in the number of haplotypes and variants in the full reference panel. For RAM and speed reasons..." be rewritten to "Given the two-part iterative process, QUILT1 has linear time complexity dependent on the number of haplotypes and variants in the full reference panel. To minimize RAM and runtime..."

Your suggested alternate sentence has a different meaning to the original sentence. It's not the iterative process itself, but rather specifically the second part, that yields linear computational complexity in the number of reference haplotypes (this is the key bit, both the first and second part yield linear time complexity in the number of SNPs). We have left this sentence unchanged, as we think it's less of a problem compared to the others you highlighted.

2. Including the number of SNPs as another factor in the runtime / memory usage could be interesting. For example, subset the SNPs in the lc-WGS data and see how it varies with the number of SNPs included. There is some analysis on this for the QUILT-nipt method vs QUILT2, though it only focuses on all variants vs only common, and does not mention the number of SNPs included at these thresholds.

Thank you for the comment. We have partially done this, in that in Supplementary Figure 5, we evaluated the runtime and memory by adjusting a threshold which determines at which frequency SNPs are considered common vs rare in QUILT2. More generally, usually, the number of SNPs in a panel is a more minor concern than the number of haplotypes. In Figure 3, where we show the scaling properties of the methods, the number of SNPs in the panels varies as well, as monomorphic SNPs are removed from smaller panels. So in that sense, we do evaluate how the number of SNPs effects run time, and we don't think another dedicated analysis would be particularly fruitful for readers of the manuscript.

3. Given that low-coverage WGS imputation methods often differ from GWAS array imputation methods in that the former relies on genotype probabilities whereas the latter relies on genotype calls, is QUILT2 incompatible with GWAS array data? This point is worth clarifying to ensure use cases are clear to readers.

QUILT2 as written uses sequencing reads directly. It could be modified to use array genotype information, but there are many dedicated methods already to do this. We note that QUILT2 operating on sequencing reads is a distinct advantage of the method, as we've shown in the paper, e.g. with the long read analysis, in contrast with GLIMPSE2, which only uses input information in a per-SNP manner.

We have amended the Introduction in general, and have included this sentence below. Implied but not stated is that QUILT cannot use array input. We can make this more explicit if the reviewer would like us to.

QUILT and GLIMPSE further differ in how they treat input data, with GLIMPSE using per-SNP genotype likelihoods, while QUILT uses and phases the sequencing reads directly, which is preferable for long-read data.

4. Common variant definition of $MAF=0.1-0.2$ seems a little funny. Why not just $MAF > 0.1$?

We stratified variants into bins based on minor allele frequency, as is standard in methods papers like these, as variant frequency is strongly correlated with imputation. This is most important for variants at low frequency. We were perhaps overzealous in our bin choices for common variants. Nonetheless more bins at higher frequencies makes for slightly nicer figures, and reporting this way ensures consistency between figures and text. If the reviewer remains keen we can change this e.g. to 5 or 10 to 50%, but it is unlikely to affect a qualitative interpretation of the results. For reference, we used 20-50% for the QUILT1 paper.

5. Given that QUILT2 imputed more accurately than GLIMPSE2 with UKB WGS but not phase 3 1kGP and HRC, is there better genotype error tolerance in GLIMPSE2? Notably, a 30x WGS version of 1kGP is available.

Thank you for the comment. Yes we do use the higher coverage 30X 1kGP panel. It is tricky to know for sure what's driving the differences, which are minimal. You're right that it might have to do with better genotype error tolerance and how this relates to the PBWT based heuristics.

6. In Figure 2a, it is hard to tell the coverage lines apart because points overlap. The authors might consider using a color gradient for increasing coverage, with QUILT2 vs GLIMPSE instead indicated by line type (e.g. dashed vs solid). Comparing accuracy across Figure 2a, 2b, and 2c is challenging as the scale is changing.

Thank you for the comments. We now re-designed the plot and added 4x in the benchmarking as suggested by another reviewer. We hope this will facilitate interpretation.

7. Very interesting to see the ancient DNA imputation results. Though in Figure 2d (and corresponding supplemental figures) - is there a particular reason to switch from MAF as x-axis (for figures 2a-c) to coverage as x-axis and color as MAF in Fig2d? Could consider switching the axis to match the other plots, though not critical to do this.

Thank you for the comment. We agree this was confusing. We have re-designed it now to be more in line with the other figures.

8. Non-reference concordance is a bit more of a forgiving metric than aggregate r^2 . Do the authors care to comment on the relevance of this in the comparison of aDNA vs modern samples?

We agree, but here we are following the style in the GLIMPSE2 paper and other papers on ancient DNA.

9. In Figure 3, the legends are a bit confusing, since those shown in A differ from in B but are relevant to A-C. I would suggest extracting a common legend to the bottom of this figure, then reducing the y-axis for panel b, as it is hard to distinguish between lines. The description of memory usage in the text describing this figure could be clearer about the relative comparison between QUILT2 and GLIMPSE2 (note that GLIMPSE2 is not mentioned here): "We note that QUILT2 uses slightly higher RAM given 100 samples to run, and QUILT2 also runs faster with lower RAM for lower coverage data (Figure 3C, Supplementary Figure 5, 6)."

We appreciate this suggestion. We have re-designed the figure now. In addition, we have changed the text to be clearer about the relative comparison of RAM usage with sample size for GLIMPSE2 and QUILT2.

RAM usage is approximately constant for GLIMPSE2 with sample size, whereas it increases in QUILT2 (Figure 3). In addition, QUILT2 also runs faster for lower coverage data (Supplementary Figure 5, 6).

We do note that both methods have reasonably low RAM usage, which we don't estimate to be a hindrance for most users.

10. The NIPT comparisons in Figure 4 are nice. The authors might want to comment on whether imputing only common fetal variants is the best strategy given the focus on embryonic screening for

structural variants and potentially Mendelian variants, or whether lcWGS is an uncommon current application but with approaches like QUILT2 enabling more.

Thank you for the comment. Sequencing cfDNA and NIPT has been proven to be the most accurate method for aneuploidy detection, and it is increasingly cheap and easy to perform. Specifically NIPT using genome-wide lcWGS, vs targeted sequencing, has been adopted at scale with many millions of samples already performed in both the Netherlands and China. We hope that, with this paper, people will also see the benefits of imputing the mother and the fetus for both research and clinical opportunities. We also hope that, as we've shown, accuracy improves as sequencing coverage improves, and in such cases, we expect to see further improvement in the ability to genotype Mendelian (rare) variants. Further, for structural and copy number variants, we've mentioned this as an area for further improvement in the Discussion. Finally for rare variants, imputing these, at low coverage, is very difficult, and likely insufficient for clinical purposes, hence why in the Discussion we highlight alternative approaches like non invasive fetal exome sequencing. We think we've covered these issues in the Discussion in the second and third paragraphs though if the reviewer feels we could do more to address them we are happy to add more text in.

11. Any thoughts on why there is a downward trend in aggregated R2 for common SNPs in Fig 4, both maternal and fetal genomes? More pronounced for diploid than NIPT and more pronounced with increasing coverage.

Thank you for the comments, which are very good points. The aggregated R2 starts dropping towards high frequency variants can be explained by several factors. First, for high frequency variants, it is more likely that the paternal alleles/reads are different to the maternal allele/reads, which will be more pronounced given high depth sequencing. Then, when the paternal alleles are different to the maternal alleles, the ability to distinguish them in the model explains how the results will look like. With the diploid model, where the paternal origin is wrongly modeled as either the maternal transmitted or maternal untransmitted reads, it just can not distinguish them and hence will be confused. As an example, given the truth allele of maternal untransmitted, maternal transmitted and paternal origin is 1, 1, and 0 respectively (hence the mother is 1|1 homozygous alternatives), if the fold difference between maternal untransmitted (1) and paternal origin (0) is very low, say 2-fold (means fetal fraction is very high), then there is 1 / 3 chance to call the mother 1|0, which is wrong. In contrast, if the fetal fraction is very low, say fold change in the coverage of maternal untransmitted vs paternal origin is 9-fold, then there is only 1/10 chance calling the mother genotype wrong. This downward trend in the diploid model was also observed by other studies, e.g the recent one <https://www.sciencedirect.com/science/article/pii/S2211124724011501#sec3>.

Now, for the nipt mode, relatedly, even though we model the paternal origin reads as a third distinct haplotype, the higher fetal fraction can still blur the distinction between the paternal haplotype and the maternal untransmitted haplotype, making it easier for the model to confuse them e.g. at ff = 10%, we expect 45% maternal untransmitted vs 5% paternal origin reads, a nine-fold difference, while at ff = 30%, we expect 35% mat-u vs 15% pat, around 2-fold difference. As an example, given the truth genotype of mother, father and fetus is 0|1, 1|1 and 1|1 respectively, if the fold change between maternal untransmitted (0) and paternal origin (1) is only 2 compared to 9 (when fetal fraction is low), then it challenges the model more in distinguishing them, hence it might be a higher chance to call the fetus as 0|1 instead of 1|1.

12. Why were only quantitative traits used for the GWAS / PRS analyses? Do the authors expect a change in behavior for logistic mixed models? Don't necessarily have to test binary traits, but just include a sentence explaining why quantitative should behave the same as binary traits.

We expect differences in power between array vs lcWGS for quantitative traits to be similar for binary ones. We chose to use only quantitative traits so that all traits could be analysed in the same way. We have added a sentence to the results to make it clearer that we expect binary traits to show broadly the same results

We only tested quantitative traits but expect binary traits to show broadly similar results.

13. For the NIPT GWAS locus identification part, the authors use a loose definition of $p < 0.05$ to describe identification. Figure 5B shows that the vast majority are identified with nominal significance, so a concordance check on effect size (same direction at least?) may also be warranted. It might be helpful to reorder these bars so that the mother array followed by mother with decreasing coverage are shown, followed by the simulated fetus, as it's otherwise hard to compare the relative change in performance. Blue and orange coloring in Figure 5B should be switched to not be confused with the blue/orange for fetus/mother color scheme in other plots. In terms of relative gains in discoverability expected across technologies, this publication might be worth discussing: Gaynor, S.M., Joseph, T., Bai, X., Zou, Y., Boutkov, B., Maxwell, E.K., Delaneau, O., Hofmeister, R.J., Krasheninina, O., Balasubramanian, S., et al. (2024). Yield of genetic association signals from genomes, exomes and imputation in the UK Biobank. *Nat. Genet.* 56, 2345–2351.

Thank you for the comments. First, thank you for suggesting a better way to present the results in Figure 5. We've now re-designed it as your suggestion. Second, yes, a p-value threshold of 0.05 is lenient, however the main purpose of this is as a means to show that results are generally consistent when doing using arrays, or lc-WGS using NIPT. As an alternative to checking concordance of effect size direction, below we check the Pearson correlation between effect sizes versus arrays, and we see they are incredibly similar, ranging from 0.998, to 0.935. As such, any similarity in the number of loci shown in Figure 5B is undoubtedly from the high similarity in underlying genotypes.

14. Given the focus on NIPT, how many common clinically actionable HCMG genes or other common variants relevant to fetal Mendelian disease (e.g. CFTR deltaPhe508) are well-imputed? Is low-coverage WGS the best strategy here, or might a WEGS or blended strategy that enriches for higher coverage at interpretable coding variants be worth discussing?

Thank you for the comments. In the paper, we didn't look into the specific common variants that are relevant to fetal Mendelian disease. Assuming for example the CFTR deltaPhe508 variant shows similar imputation accuracy as for other variants at similar frequency (1.5% in Europeans by gnomAD), then given 1x, 2x, 4x and 10x NIPT lcWGS with 10% FF, the imputation accuracy r^2 for the fetal genotype would be around 0.713, 0.809, 0.866, and 0.885 respectively. This may be sufficient to request confirmatory sequencing, or in the context of pre-existing knowledge of being a carrier of the variant, but in itself does not meet reasonable diagnostic thresholds. As such, we've modified our discussion to emphasize this, and the value of recent development of non-invasive fetal exome sequencing.

Nonetheless, the imputation of specific variants, in particular rare variants, remains challenging. Therefore it is unlikely to be used for clinical diagnosis, and in that context, recent approaches using non-invasive deep fetal exome sequencing are more promising 22,23. We note that further extension to QUILT2-nipt should still operate on fetal exome sequencing, so in addition to accurate rare variants, such data would also allow for genotype imputation and PRS calculation.

15. The discussion point about the clinical actionability of de novo CNVs e.g. 22q11.2 is interesting but also confusing, as imputation methods like QUILT2 use a reference panel to only impute existing

variants in the population. This is framed as a potential future development opportunity for QUILT2 that seems highly speculative at present as a class of variation undetectable by any imputation method.

People have previously called CNVs including de novo ones using array data e.g. PennCNV. People have also looked at calling more complicated chromosomal aberrations using array data from blood e.g. work by Po-Ru Loh and colleagues using the UKB. That being said, QUILT2 is one of the very few imputation methods that works directly on sequencing reads, rather than genotype probabilities / genotype likelihoods. With QUILT2, Gibbs sampling on read labels yields read label estimates that directly assign reads to haplotypes. As such, it is also a read phasing method. We have a new paper in preparation describing improved SV genotyping accuracy by using the per-read phasing information already output from QUILT2, as this is additional evidence, beyond what current methods use, to generate a likelihood for whether an SV call is correct or not.

In the future, a simple modification to the QUILT2 model could be developed to consider a “copy number”. For recurrent microdeletions and duplications, with known start and end points, this would not be difficult, as the currently used prior probability on read labels (based on fetal fraction) would be modified to include copy number, given known start and end points. One could then for instance compare the likelihood of a normal model (with normal copy number) to other models (e.g. paternal origin duplication between LCR A and LCR D in 22q11), and “call” the event if a sufficient difference in likelihood were present. For general de novo CNVs, more complicated future development that could adopt varying copy number (e.g. as a hidden variable itself) would permit a general purpose solution, however that would be more complicated to implement. Even further in the future there could be scope to implement a read phasing and local realignment model, like used for variant calling in Platypus and other more modern variant callers e.g. the GATK HaplotypeCaller.

As such we think it’s a reasonable speculation to motivate interest in this result and future work based on this method or using these ideas for future methods.

16. Methods: “ λ is the parameters of our model” - this parameter needs to be defined more clearly.

We have changed this to

λ is the parameters of our model, including the recombination rate and reference haplotypes

We also mention later another parameter “FF is the fetal fraction, which is required as an input and can be obtained from NIPT report,”

17. Line 623: “Real big” reference panel - change to “very large” or “one of the largest”

Thank you, we have corrected the language.

18. Not sure if it’s the pdf formatting, but for the notation in the supplemental note, there is no semi-colon.

Thank you for this. We wrote “semi-colon” but we meant colon. We have corrected this in the supplement.

Reviewer #4 (Remarks on code availability):

The README provides detailed instructions, including conda environment installation, test run information, and tutorial pages for the various use cases of QUILT2. A number of issues have been opened on the github, which is normal with any bioinformatics software. Importantly, the authors seem responsive to those ongoing issues. We did not try to install the software, but based on the github repository, the usability seems high.

Thank you for your kind comments.

Reviewer #5 (Remarks to the Author):

R1

I would like to add a further comment regarding large-scale GWAS analysis. I understand the rationale for splitting jobs across samples, which provides a runtime advantage for QUILT2 when dealing with small sample sizes. In your response to Reviewer 3 regarding large scale GWAS analysis, you note: “As such, the issue of running large datasets with many thousands of samples is really just an issue of how to most efficiently run per-sample, which is why we’ve focused as we have on comparatively small sample sizes like shown in Figure 3.”

If I understand correctly, the authors suggest splitting large datasets both into chunks and by sample. My concern is that this approach introduces substantial overhead in starting and managing multiple machines, as it generates a large number of tasks per imputation job, potentially offsetting any runtime advantage. While fast per-sample processing can be beneficial, higher parallelisation does not make it automatically faster in a cloud environment (though it could be advantageous on a local Slurm cluster). I am curious whether the authors could provide an overall estimate of cloud-based imputation performance (e.g., using a larger sample size for both methods -> runtime, costs). By comparison, current state-of-the-art imputation servers using microarray data typically use chunking only, which works efficiently in the cloud. Even when using spot instances, which may be interrupted more frequently, the cost advantage is substantial (I think up to 90% cheaper than on-demand instances). I would also appreciate clarification on whether the Snakemake pipeline supports per-sample chunking, as this is not as straightforward for end-users as region-based chunking (due to splitting and merging in correct order). Finally, I am unfamiliar with the distinction between “high” and “low” priority jobs. In the UK Biobank instance rating card, the main difference is between spot and on-demand instances (0.2336 vs 1.2832 GBP/hour for the instance type used), not sure if these terms can be used interchangeably

re: estimate of cloud-based imputation performance: We have presented what we think is a reasonable strategy for imputing samples, and this costing includes the overhead cost of starting VMs. In the UKB environment, we suggest running a reasonably similar number of samples (N = 32) as the GLIMPSE2 authors (N = 100) samples. This costing and setup reflects what would be most important, the total cost, when running a large number of samples. It is certainly true that in different computer architectures, including cloud vs academic, as well as different sample size setups, e.g. running very few samples, a different split between sample chunking and region chunking might make more sense. But overall we think we’ve presented a comparison that accurately reflects the main issues that users would experience and is fair to both QUILT2 as well as GLIMPSE2.

re: Snakemake: the Snakemake pipeline does not currently perform per-sample chunking. However this is generally easier to do compared to per-region chunking. Since the imputed SNPs will be the same for all samples, merging VCFs afterwards is easy using tools like the GATK MergeVCFs or BCFTOOLS merge.

Re “high” vs “low”: When submitting a job, the VM is assigned depending on the specified priority, which takes three values: high, normal or low. High priority jobs will use on demand instances, while low priority jobs will use spot instances. A normal priority job running on a spot VM can be interrupted and after several restarts it will turn into a high priority job running on an on-demand VM. More on this

<https://dnanexus.gitbook.io/uk-biobank-rap/administrator/managing-usage-and-storage-costs> and <https://dnanexus.gitbook.io/uk-biobank-rap/working-on-the-research-analysis-platform/managing-jobs/managing-job-priority>.

R2

Thanks to the authors for addressing my comments.

Given their reply and that they only showed that their approach partly may work in an NIPT context (i.e. only one sample). If the editor decides to publish this manuscript, I recommend to remove NIPT (fetal cell-free DNA) in the abstract and the title and instead limit their claims as potential application of QUILT2 in an NIPT context. The low imputation capability and testing QUILT2 on only one actual NIPT sample, is far from reality.

In my opinion, QUILT2 is far from "enabling imputation of the maternal and fetal genome using cell free non-invasive prenatal testing (NIPT) data."

Given the editorial need to make the title more succinct, as well as a need to remove punctuation from the title, we have removed the specific reference to cfDNA NIPT imputation in the title. However we have left it in the abstract, as this is a major feature of this work. QUILT2 is the only method that performs imputation on cfDNA NIPT data, and allows for imputation of the mother and the fetus both, and also improves the accuracy of imputation for both, with the accuracy of the maternal imputation being quite high. As NIPT sequencing is performed millions of times a year, we think it is important to highlight this part of QUILT2 in the abstract.